# Forebrain deletion of the dystonia protein torsinA causes dystonic-like movements and loss of striatal cholinergic neurons

Samuel S Pappas[1]*, Katherine Darr[1], Sandra M Holley[2], Carlos Cepeda[2], Omar S Mabrouk[3], Jenny-Marie T Wong[4], Tessa M LeWitt[1], Reema Paudel[5], Henry Houlden[5], Robert T Kennedy[4], Michael S Levine[2], William T Dauer[1,6]*

[1]Department of Neurology, University of Michigan, Ann Arbor, United States; [2]Intellectual and Developmental Disabilities Research Center, Brain Research Institute, Semel Institute for Neuroscience, David Geffen School of Medicine, University of California, Los Angeles, Los Angeles, United States; [3]Department of Pharmacology, University of Michigan, Ann Arbor, United States; [4]Department of Chemistry, University of Michigan, Ann Arbor, United States; [5]Department of Molecular Neuroscience, Institute of Neurology, University College London, London, United Kingdom; [6]Department of Cell and Developmental Biology, University of Michigan, Ann Arbor, United States

**Abstract** Striatal dysfunction plays an important role in dystonia, but the striatal cell types that contribute to abnormal movements are poorly defined. We demonstrate that conditional deletion of the DYT1 dystonia protein torsinA in embryonic progenitors of forebrain cholinergic and GABAergic neurons causes dystonic-like twisting movements that emerge during juvenile CNS maturation. The onset of these movements coincides with selective degeneration of dorsal striatal large cholinergic interneurons (LCI), and surviving LCI exhibit morphological, electrophysiological, and connectivity abnormalities. Consistent with the importance of this LCI pathology, murine dystonic-like movements are reduced significantly with an antimuscarinic agent used clinically, and we identify cholinergic abnormalities in postmortem striatal tissue from DYT1 dystonia patients. These findings demonstrate that dorsal LCI have a unique requirement for torsinA function during striatal maturation, and link abnormalities of these cells to dystonic-like movements in an overtly symptomatic animal model.

*For correspondence: samuelpa@med.umich.edu (SSP); dauer@med.umich.edu (WTD)

**Competing interests:** The authors declare that no competing interests exist.

## Introduction

Primary dystonia encompasses a group of sporadic and inherited disorders characterized by disabling involuntary twisting movements. This sole clinical feature of primary dystonia implies selective abnormalities of motor pathways. The absence of an overt neuropathological signature complicates identification of pathogenic brain structures and cell types, and underlies the widely accepted notion that primary dystonia results from abnormal functioning of a structurally intact central nervous system (CNS).

Several lines of evidence implicate the striatum as the major node of dysfunction in primary dystonia. *Secondary* dystonia–where symptoms result from CNS damage or exogenous pharmaco-logical insult–is linked strongly to striatal (especially putaminal) damage (*Marsden et al., 1985*). Therapeutic benefit from antimuscarinic drugs (*Burke et al., 1986*) also implicates striatal dysfunction, as striatal cholinergic interneurons play a poorly understood but important role in motor control. Striatal-associated behavioral (*Carbon et al., 2011*) and functional imaging abnormalities are present

**eLife digest** Dystonia is disorder of the nervous system that causes people to suffer from abnormal and involuntary twisting movements. These movements are triggered, in part, by irregularities in a part of the brain called the striatum. The most common view among researchers is that dystonia is caused by abnormal activity in an otherwise structurally normal nervous system. But, recent findings indicate that the degeneration of small populations of nerve cells in the brain may be important. The striatum is made up of several different types of nerve cells, but it is poorly understood which of these are affected in dystonia.

One type of dystonia, which most often occurs in children, is caused by a defect in a protein called torsinA. Pappas et al. have now discovered that deleting the gene for torsinA from particular populations of nerve cells in the brains of mice (including a population in the striatum) causes abnormal twisting movements. Like people with dystonia, these mice developed the abnormal movements as juveniles, and the movements were suppressed with 'anti-cholinergic' medications. Pappas et al. then analyzed brain tissue from these mice and revealed that the twisting movements began at the same time that a single type of cell in the striatum—called 'cholinergic interneurons'—degenerated. Postmortem studies of brain tissue from dystonia patients also revealed abnormalities of these neurons.

Together these findings challenge the notion that dystonia occurs in a structurally normal nervous system and reveal that cholinergic interneurons in the striatum specifically require torsinA to survive. Following on from this work, the next challenges are to identify what causes the selective loss of cholinergic interneurons, and to investigate how this cell loss affects the activity within the striatum.

in primary dystonia (reviewed in *Pappas et al. (2014)*), and altering basal ganglia output with deep brain stimulation therapy is an effective dystonia treatment (*Vidailhet et al., 2013*). Despite this evidence, the key striatal cell type(s) that drive dystonic movements are unknown.

Studies aimed at defining mechanistic features in primary dystonia primarily use rodent models of DYT1 dystonia, a neurodevelopmental disorder manifesting during childhood, and the most common inherited primary dystonia. DYT1 dystonia is caused by a dominantly inherited mutation of the *TOR1A* gene that impairs function of the encoded protein torsinA. TorsinA is an endoplasmic reticulum/ nuclear envelope-localized AAA+ ATPase (*Ozelius et al., 1997*) implicated in protein quality control and nuclear membrane-localized functions (reviewed in *Dauer (2014)*). Heterozygous $Tor1a^{\Delta E/+}$ mice (mimicking the human DYT1 genotype) do not exhibit any overt abnormalities, while constitutive *Tor1a* knockout and homozygous ΔE knock-in mice both exhibit perinatal lethality (*Goodchild et al., 2005*; *Tanabe et al., 2012*). Transgenic mice overexpressing wild type or mutant torsinA do not exhibit overt motor abnormalities (*Sharma et al., 2005*), but are used to explore striatal electrophysiological abnormalities linked to overexpression of mutant torsinA. These studies demonstrate that in DYT1 mutant transgenics, striatal large cholinergic interneurons (LCI) exhibit a paradoxical response to dopamine D2 receptor agonists that may be involved in abnormalities of corticostriatal plasticity (*Pisani et al., 2006*; *Martella et al., 2009*; *Sciamanna et al., 2011*; *Grundmann et al., 2012*; *Sciamanna et al., 2012a*, *2012b*). The relationship of these abnormalities to dystonic movements is unclear, as they occur in rodent models both with and without abnormal movements.

Conditional deletion of torsinA in single brain regions (e.g., cortex, striatum) or cell types (e.g., cerebellar Purkinje cells, cholinergic neurons) implicated in the disease causes subtle changes in motor function, but no overt abnormal movements (*Yokoi et al., 2008*, *2011*; *Zhang et al., 2011*; *Sciamanna et al., 2012a*). Overt twisting movements are only observed in DYT1 model mice where torsinA function is impaired in precursor cells giving rise to multiple neuronal cell types (*Liang et al., 2014*). These results implicate the importance of developmental timing of torsinA loss of function and the potential involvement of multiple dysfunctional cell types in disease pathophysiology. These models exhibit focal neurodegeneration in a discrete set of sensorimotor structures, and together with human subject neuroimaging studies (reviewed in *Ramdhani and Simonyan (2013)*), raise questions regarding the 'normal structure, abnormal function' hypothesis of primary dystonias.

To further explore this structure-function question as well as the potentially important role for torsinA during the early development of corticostriatal circuitry, we developed a novel mouse model

by deleting torsinA with *Dlx5/6-Cre*, which acts in progenitors of forebrain GABAergic and cholinergic neurons (*Monory et al., 2006*). This model exhibits face, construct and predictive validity. These mice are initially normal, but exhibit overt motor deficits as juveniles, coincident with selective loss of striatal LCIs and related electrophysiological abnormalities. Similar to DYT1 patients, the abnormal twisting and clasping movements of these mice are reduced significantly with chronic antimuscarinic administration. Moreover, we identify cholinergic abnormalities in postmortem putamen from DYT1 subjects. These observations are the first to demonstrate the unique vulnerability of a specific striatal cell type to torsinA loss of function, and have important implications for the understanding of disease pathogenesis and the development of targeted therapeutics.

## Results

### Conditional deletion of TorsinA from forebrain cholinergic and GABAergic neurons causes motor abnormalities during juvenile CNS maturation

We conditionally deleted *Tor1a* from precursors of forebrain GABAergic and cholinergic neurons by crossing *Dlx5/6-Cre* and *Tor1a* 'floxed' mice (*Monory et al., 2006*; *Liang et al., 2014*). Using mT/mG and Rosa26 LacZ Cre-reporter lines (*Soriano, 1999*; *Muzumdar et al., 2007*), we confirmed that Cre activity was restricted to forebrain structures (striatum, cortex, globus pallidus, basal forebrain, reticular thalamic nucleus), and included both direct and indirect pathway-projecting striatal neurons (*Figure 1A*). TorsinA immunohistochemistry confirmed the essentially complete deletion of torsinA protein from striatum, partial deletion from cortex (reflecting loss from GABAergic interneurons), and sparing of the thalamus–with the exception of the inhibitory neurons of the reticular thalamic nucleus (*Figure 1B*). *Dlx5/6-Cre⁺;Tor1aᶠˡˣ/⁻* mice (herein Dlx5/6 conditional KO 'Dlx-CKO') are born in the expected Mendelian ratio and are indistinguishable initially from littermate controls, including normal postnatal weight gain (*Figure 1—figure supplement 1*).

Nissl-stained brain sections of Dlx-CKO mice did not demonstrate gross or microscopic abnormalities of forebrain architecture, and immunostaining showed no evidence of reactive gliosis or neural injury (*Figure 1C*). Cortex and striatal development was normal, as assessed by size measurements throughout development (*Figure 1D,E*). Dlx-CKO pre-weaning motor function did not differ from littermate controls (*Figure 1F*). These data indicate that initial development and postnatal maturation of forebrain motor circuitry occurs normally in the absence of torsinA in forebrain cholinergic and GABAergic neurons.

Dystonia in humans is commonly exacerbated by action, and may occur exclusively in particular settings or during specific motor tasks (e.g., runner's dystonia, dystonic writer's cramp). We assessed motor function during gait and during tail suspension, when mice vigorously kick their limbs and attempt to attain an upright body posture. Gait analysis was largely unremarkable (*Figure 1—figure supplement 2*). Dlx-CKO mice exhibited normal behavior during tail suspension up to 14 days of age, but nearly 100% of the animals developed severe forelimb and hindlimb clasping behaviors beginning at 15–17 days of age (*Figure 1G*). This abnormal behavior remained fixed for the duration of the animal's life, to at least 1 year of age (17 out of 18 Dlx-CKO mice clasped during tail suspension at 1 year). A subset of these mutants simultaneously developed severe abnormal twisting of the trunk (~70% of mice; *Figure 1H*). Dlx-CKO mice also developed a defect in the ability to hang from a wire grid (*Figure 1I*, upper panel) that in some cases appeared related to abnormal hindpaw twisting. In contrast, motor learning and gross coordination appeared normal, as assessed by the ability to remain on an accelerating rotarod at 8 weeks of age (*Figure 1I*, lower panel). Dlx-CKO mice are also significantly hyperactive in the open field (*Figure 1—figure supplement 3*). These observations demonstrate that torsinA loss of function in forebrain GABAergic and cholinergic neurons is sufficient to cause action-induced abnormal twisting movements. The onset of these abnormal movements during juvenile CNS maturation and their persistence into adulthood broadly resembles the natural history and symptomatology of DYT1 dystonia (*Dauer, 2014*).

### Anticholinergic treatment ameliorates abnormal twisting of Dlx-CKO mice

Chronic antimuscarinic administration is a common therapy for DYT1 dystonia (*Burke et al., 1986*), prompting us to evaluate the ability of antimuscarinics to ameliorate the abnormal twisting movements in Dlx-CKO mice. Dlx-CKO mice were treated with once-daily injections of the antimuscarinic scopolamine (5 mg/kg, s.c.) or saline for 10 days. The duration of forelimb clasping,

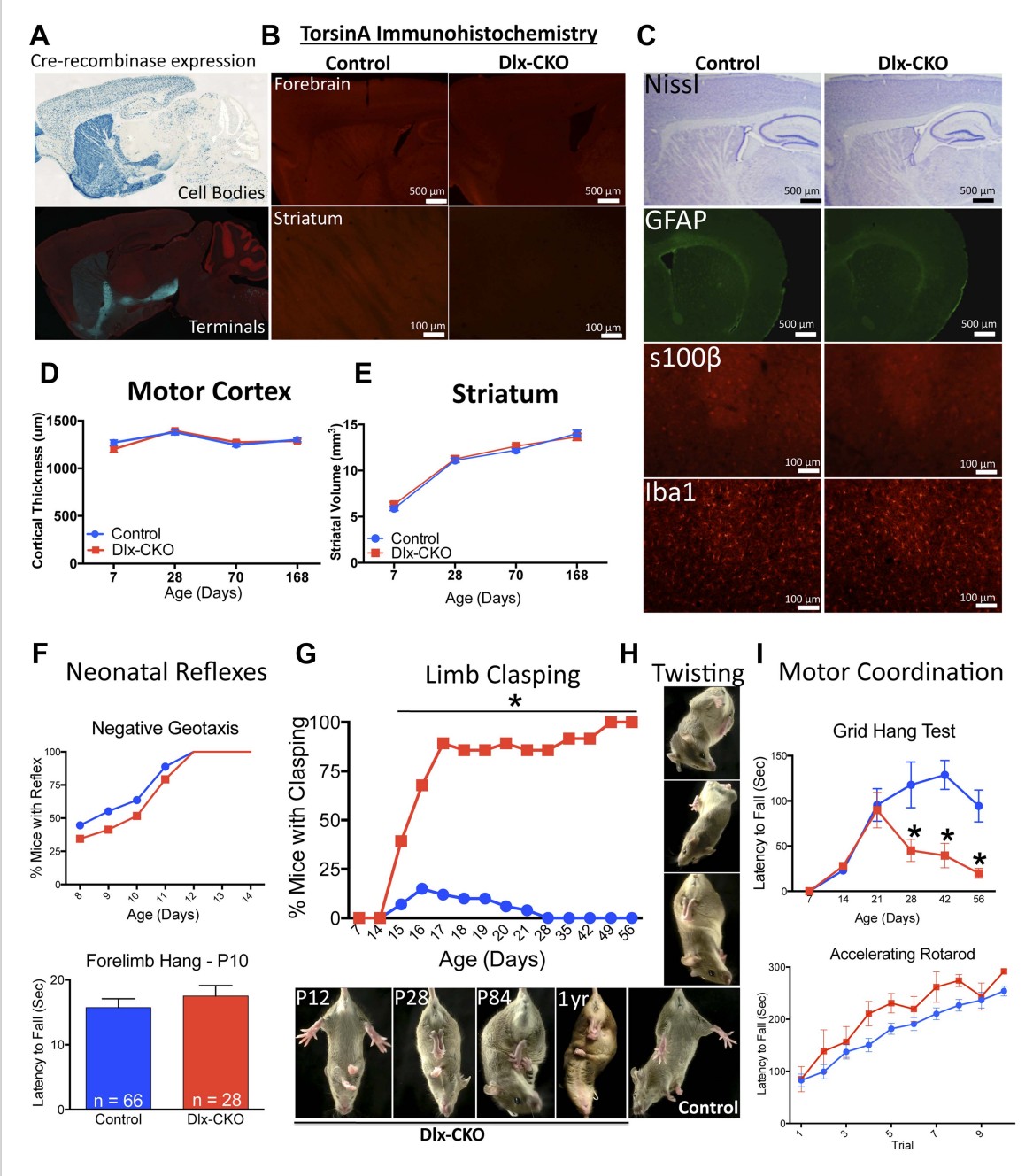

**Figure 1**. Conditional TorsinA deletion from forebrain GABAergic and cholinergic neurons causes dystonic-like movements in juvenile mice. (**A**) *Dlx5/6-Cre* expression is restricted to forebrain, as demonstrated by rosa26 LacZ and mT/mG reporter lines. (**B**) TorsinA immunohistochemistry demonstrates complete torsinA deletion in the striatum and partial deletion in the cortex. (**C**) Dlx-CKO mouse forebrain architecture appears normal (Nissl) and there is no evidence of gliosis (GFAP, s100β, Iba-1). (**D–E**) Gross striatal and cortical development appears normal. Cortical thickness: two-way ANOVA main effect of age $F_{3,65} = 17.24$; $p < 0.0001$, genotype $F_{1,65} = 0.35$; $p = 0.55$); striatal volume: main effect of age $F_{3,65} = 307.0$; $p < 0.0001$; genotype $F_{1,65} = 0.724$; $p = 0.39$. (**F**) The behavior of neonatal Dlx-CKO mice is normal. Negative geotaxis and forelimb suspension did not differ from littermate controls. Forelimb suspension: t-test $t_{(92)} = 0.753$; $p = 0.45$. (**G–H**) Dlx-CKO mice develop severe forelimb and hindlimb clasping at P15 (Chi square test, $X^2 = 64.03$; $p < 0.0001$), and a subset exhibits severe trunk twisting. (**I**) Dlx-CKO mice develop an inability to hang from a wire grid at 1 month of age (two-way ANOVA; main effect of genotype $F_{1,269} = 16.63$; $p < 0.0001$, time $F_{6,269} = 6.613$; $p < 0.0001$; and interaction $F_{6,269} = 2.285$; $p = 0.036$).
*Figure 1. continued on next page*

*Figure 1. Continued*

Motor learning remains intact, as demonstrated by the accelerating rotarod test (two-way ANOVA main effect of trial $F_{9,324} = 38.27$ p < 0.0001, genotype: $F_{1,36} = 3.591$; p = 0.066).

The following figure supplements are available for figure 1:

**Figure supplement 1**. Postnatal weight gain is normal in Dlx-CKO mice.

**Figure supplement 2**. Dlx-CKO mice exhibit normal motor function during gait.

**Figure supplement 3**. Dlx-CKO mice are hyperactive.

hindlimb clasping, and trunk twisting (*Figure 2A*) was assessed on videos by blinded observers. Scopolamine-treated animals exhibited significantly less limb clasping and twisting than saline-treated animals throughout the treatment period (*Figure 2B*; main effect of drug $F_{1,141} = 36.17$; p < 0.0001, Sidak's multiple comparisons test). The symptomatic improvement appeared to depend on the continued presence of scopolamine, as behavioral benefit disappeared following a 3-week washout period. The antimuscarinic trihexyphenidyl (THP) is the most commonly used agent to treat DYT1 dystonia and is clinically validated (*Burke et al., 1986*; *Jankovic, 2013*). Similar to scopolamine, THP (5 mg/kg, i.p.) significantly reduced clasping, and this effect resolved following a 3 week washout (*Figure 2C*; Main effect of drug $F_{1,82} = 46.69$; p < 0.0001, Sidak's multiple comparisons test). These data indicate that antimuscarinics effectively reduce clasping and twisting behaviors, and support the predictive validity of Dlx-CKO mice for the study of DYT1 dystonia.

## Dlx-CKO mice exhibit selective alteration of striatal cholinergic function

To assess the neural substrate of motor dysfunction in Dlx-CKO mice, we examined major markers of striatal signaling. Western blot analyses of microdissected striatum demonstrated a significant reduction in choline acetyltransferase (ChAT), but no significant alterations in glutamic acid decarboxylase (GAD67) or tyrosine hydroxylase (TH) expression (*Figure 3A,B*), suggesting a specific abnormality of cholinergic elements. Consistent with this possibility, the receptor tyrosine kinase TrkA, expressed specifically by striatal LCIs (*Sobreviela et al., 1994*), was reduced by approximately 50% (*Figure 3A,C*). In contrast, expression of the medium spiny projection neuron (MSN) marker DARPP-32 did not differ significantly between Dlx-CKO mice and littermate controls (*Figure 3A,D*).

To test if the alteration of cholinergic markers reflected abnormal cholinergic neurotransmission in vivo, we performed striatal microdialysis in awake, behaving mice (*Song et al., 2012*). Levels of extracellular acetylcholine (ACh) were significantly reduced in Dlx-CKO mice (~73% reduction from control levels, performed in the presence of the acetylcholinesterase (AChE) inhibitor neostigmine; $1070 \pm 231$ nM in control [n = 6] vs $290 \pm 33$ nM in KO [n = 8]; $t_{12} = 3.89$; p = 0.0021; *Figure 3E*). In contrast, the basal extracellular concentrations of 16 other neurotransmitters and metabolites did not differ significantly from controls in a separate microdialysis study (n = 7 per group; *Figure 3F*). Consistent with the reduction of ACh, histochemical analysis demonstrated a significant reduction in striatal AChE activity, an effect that appeared most prominent in dorsolateral striatum (*Figure 3G*, assay controls in *Figure 3—figure supplement 1*).

## Selective loss of dorsolateral striatal cholinergic interneurons in Dlx5/6-CKO mice

We previously reported a link between torsinA loss-of-function and developmental neurodegeneration (*Liang et al., 2014*). To determine if cholinergic abnormalities in Dlx-CKO mice reflect loss of LCIs, we quantified the number of LCIs in control and Dlx-CKO mice at 10 weeks of age, after all abnormal behaviors are fully established. Unbiased stereological quantification of striatal ChAT-positive neurons demonstrated 40% reduction in LCI number in Dlx-CKO compared to control mice (*Figure 4A,B*). To explore whether LCIs might be lost from a specific striatal sub-region reflecting a discrete circuit (*Alexander et al., 1986*), we subdivided the dorsal striatum into four quadrants and examined cell density throughout its rostro-caudal extent (*Figure 4C*). Cell loss was non-uniform, showing a clear predilection for dorsolateral (motor) striatum, and a rostro-caudal gradient of cell loss, with relative

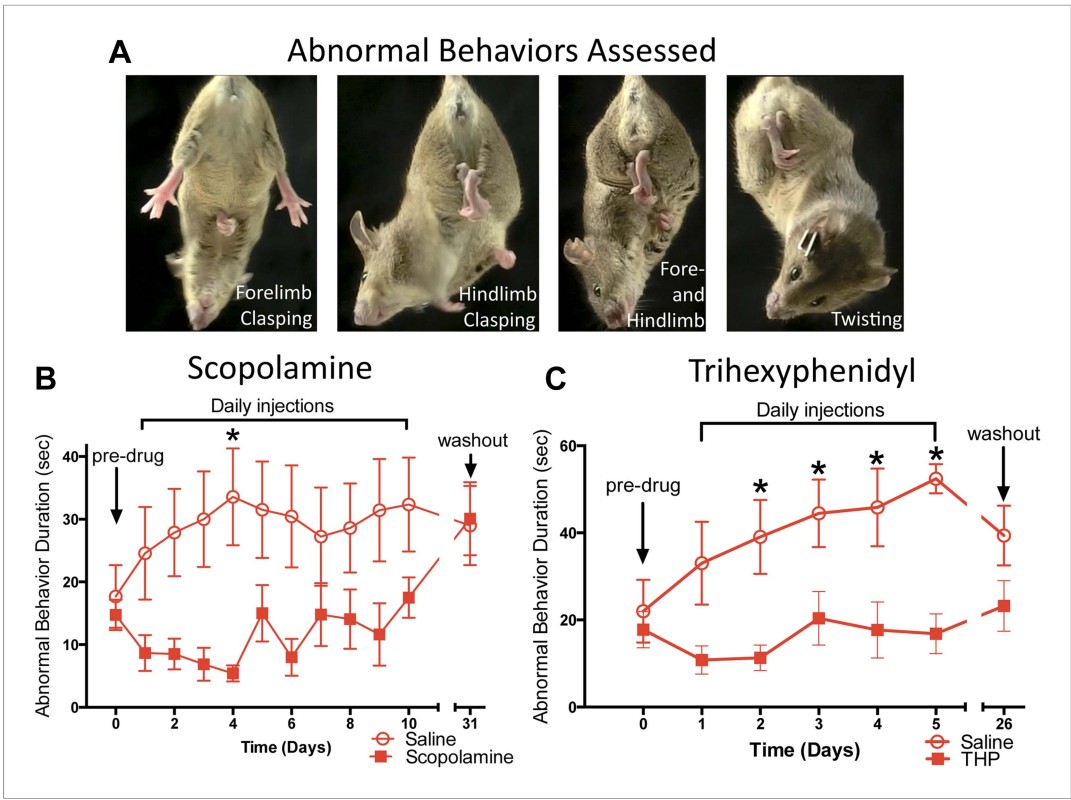

**Figure 2**. Antimuscarinic drugs ameliorate clasping and twisting behaviors in Dlx-CKO mice. (**A**) Examples of forelimb clasping, hindlimb clasping, and trunk twisting that were evaluated during review of the videos by blinded raters. (**B**) Duration of clasping and twisting was significantly reduced by once-daily 5 mg/kg scopolamine administration (tail suspension recorded 45 min following drug treatment; two-way ANOVA: main effect of drug $F_{1,141} = 36.14$; $p < 0.0001$, Sidak's multiple comparisons test. n = 8 saline, n = 6 scopolamine. This study was also repeated in a second cohort). (**C**) Clasping and twisting duration was reduced by once-daily 5 mg/kg THP administration compared to saline-treated mice (tail suspension recorded 45 min following drug treatment; two-way ANOVA main effect of drug $F_{1,82} = 46.69$, $p < 0.0001$, Sidak's multiple comparison test. n = 6 saline, n = 8 THP).

sparing of caudal regions. Strikingly, cell loss was roughly 6 times greater in dorsolateral compared to ventromedial striatum (57% vs 9% reduction; *Figure 4C,D*; *Figure 4—figure supplement 1*). Cell loss did not appear to be selective for the patch or matrix striatal subregions.

To address the possibility that the striatal abnormality reflects ChAT downregulation, we used stereology to quantify an independent marker of LCIs, vesicular acetylcholine transporter (VAChT). Dlx-CKO striata again appeared to contain 40–50% fewer VAChT+ LCIs (*Figure 4E*). To fully exclude the potential confound of phenotypic marker down regulation, we took advantage of the fact that cholinergic neurons are the largest striatal neurons, being approximately twice as large as GABAergic striatal neurons (*Kreitzer, 2009*). Stereological quantification of the number of Nissl-stained neuron profiles ≧20 μm demonstrated a ~40% reduction in Dlx-CKO compared to littermate controls (*Figure 4E*).

To examine if cell loss was specific to LCIs, we used unbiased stereology to quantify other GABAergic and cholinergic cell types from which torsinA is deleted in Dlx-CKO mice. LCIs are the only non-GABAergic neurons in the striatum, so we first quantified the number of small and medium sized Nissl-stained striatal cells (i.e., ≤20 μm), and found no significant difference in their numbers (*Figure 4F*). We next quantified well-characterized subpopulations of GABAergic interneurons in cortex and striatum. We found no significant abnormalities in the number of cortical or striatal fast-spiking (marked by parvalbumin; 'PV') or low-threshold spiking (marked by somatostatin; 'SST') interneurons (*Figure 4F,H*). Similarly, the number of striatal MSNs, (marked by DARPP-32) did not differ between 10 week-old Dlx-CKO and littermate striata (*Figure 4F*). In contrast to striatal LCIs, there is a normal number of basal forebrain cholinergic neurons (*Figure 4G*), despite the fact that these neurons also express Cre recombinase (*Sanchez-Ortiz et al., 2012*) and lack torsinA (*Figure 4—figure supplement 2*).

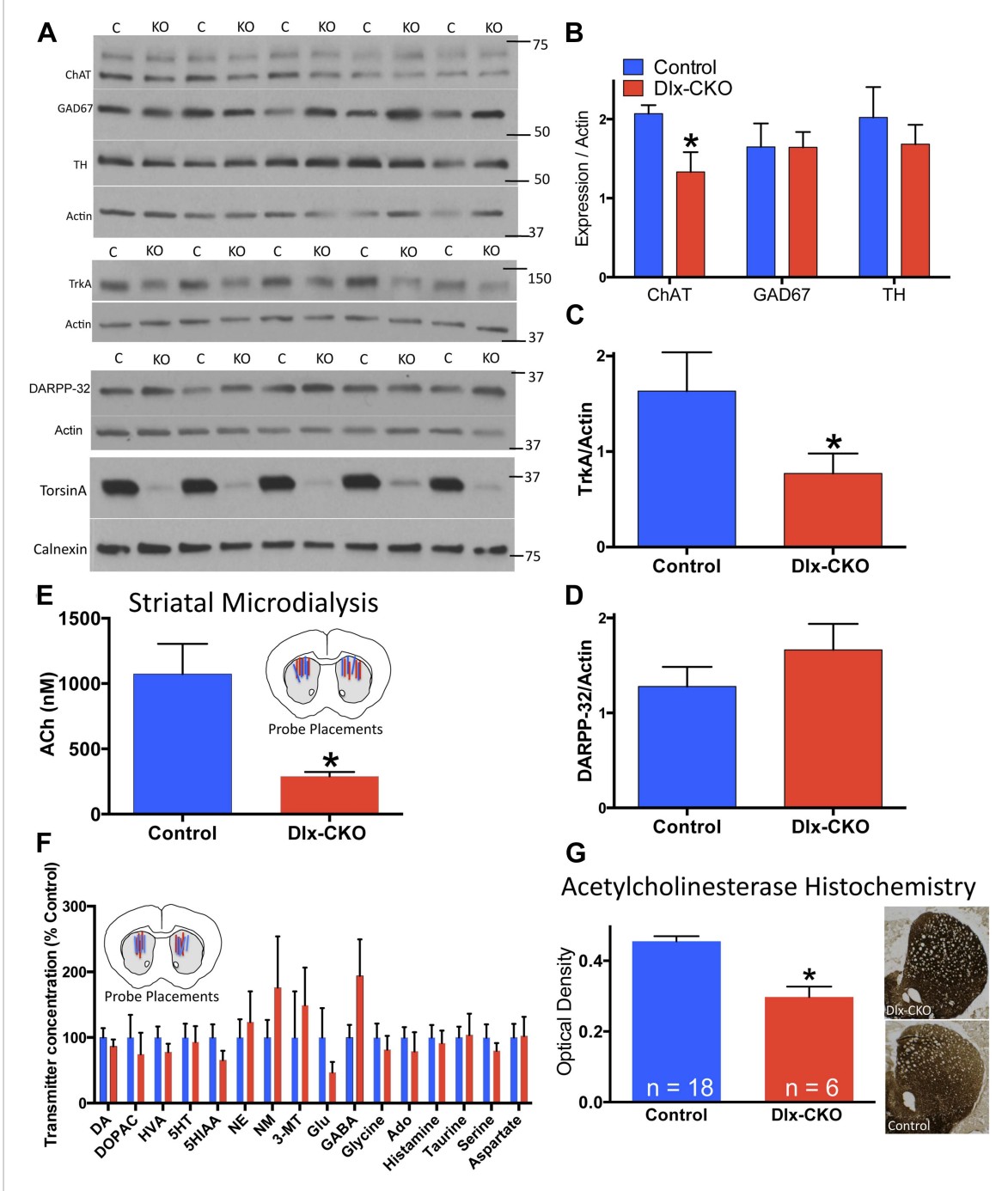

**Figure 3**. Cholinergic-specific abnormalities in the striatum of Dlx-CKO mice. (**A**) Western blots of microdissected striatum from 10 week old control and Dlx-CKO mice for markers of cholinergic, GABAergic, and dopaminergic signaling. (**B–D**) Quantification of the western blots demonstrated a selective reduction of LCI markers choline acetyltransferase (t-test: $t_{(8)} = 2.683$; $p = 0.013$) and TrkA ($t_{(8)} = 1.883$; $p = 0.048$). No differences were observed for markers of GABAergic or dopaminergic neurons (GAD67; $t_{(8)} = 0.012$; $p = 0.99$; TH; $t_{(8)} = 0.742$; $p = 0.47$; DARPP-32; $t_{(8)} = 1.12$; $p = 0.29$). (**E**) Microdialysis and HPLC-MS analysis demonstrates a significant reduction of ACh in dorsal striatum of Dlx-CKO mice (t-test: $t_{(12)} = 3.895$; $p = 0.002$; data reported as dialysate concentration and represent the average of 3 fractions per animal following neostigmine perfusion; n = 6–8 probes/group from 4 mice/group). (**F**) Microdialysis followed by benzoyl chloride derivatization and analysis by LC-MS demonstrated no significant change in any dorsal striatal neurotransmitter examined (basal values measured in absence of Acetylcholinesterase (AChE) inhibitors). Data represent the average of 5 basal collections per animal (n = 7 probes/group from 4 mice/group and are normalized to control levels (two-way ANOVA for genotype: $F_{1,190} = 0.0206$; $p = 0.88$). (**G**) AChE histochemistry on fresh frozen brain sections

*Figure 3. continued on next page*

*Figure 3. Continued*
demonstrates a significant reduction of striatal AChE in Dlx-CKO mice (t-test; $t_{(22)}$ = 5.16; p < 0.0001). Specificity of AChE reaction was confirmed using several methods (*Figure 3—figure supplement 1*).
The following figure supplement is available for figure 3:

**Figure supplement 1**. AChE histochemistry is selective for AChE.

## Striatal cholinergic neurons undergo apoptotic cell death during the onset of motor abnormalities in Dlx-CKO mice

We next explored the relationship between cholinergic cell loss and motor dysfunction. At P7, when motor function is normal (*Figure 1G,I*), there were normal numbers of ChAT+ neurons (*Figure 5A*), and normal levels of ChAT and TrkA (*Figure 5—figure supplement 1*). Progressive loss of ChAT+ neurons occurred from 1 to 2 months of age, a time period partially overlapping with the onset of motor dysfunction. Numbers of ChAT+ neurons were not reduced further at the 6-month time point (*Figure 5A*), a time when motor abnormalities similarly plateaued.

The temporal and spatial pattern of cell loss suggests that LCIs degenerate during postnatal striatal maturation. To confirm LCI cell death rather than altered cellular phenotype, we co-stained striatal sections for ChAT and cleaved caspase-3 (CC3; *Figure 5C*). We quantified the number of CC3+ and co-localized CC3/ChAT+ cells at 3 time points during striatal development. While CC3+ and CC3/ChAT+ cell numbers did not differ between mutant and littermate control mice at postnatal day 10, Dlx-CKO mice exhibited significantly more CC3/ChAT+ co-localized cells at postnatal days 12–14 and 24 (*Figure 5B*), precisely the time that abnormal movements emerge. There were no differences in the overall number of non-ChAT+ CC3 striatal cells (*Figure 5—figure supplement 2*).

## Surviving cholinergic neurons in Dlx-CKO mice exhibit morphological, electrophysiological, and connectivity abnormalities

To determine whether there are abnormalities in the remaining LCIs, which could contribute the behavioral phenotype of Dlx-CKO mice, we examined the morphological and electrophysiological properties of these cells. Surviving LCIs exhibited significant cell soma hypertrophy (*Figure 6A*). The delayed time course of this phenotype following cell loss and rightward shift of the cell size frequency histogram (*Figure 6B*) support the likelihood that surviving neurons are becoming larger. In contrast, PV+, SST+, or DARPP-32+ neurons in striatum and cortex showed no changes in soma size (*Figure 6C,D*) and MSN dendritic structure was normal, as assessed by Golgi-Cox staining and Sholl analysis (*Figure 6E–H*).

Electrophysiological analyses of surviving striatal LCIs performed after the full extent of cell loss support the possibility that abnormalities of these cells may contribute to the behavioral phenotype of Dlx-CKO mice. Spontaneous firing rates and coefficients of variation were similar for LCIs between Dlx-CKO and controls in cell-attached patch clamp mode (*Figure 7A,B*). However, cell capacitance measurements were significantly larger in Dlx-CKO LCIs than control (*Figure 7C*) when recorded at a membrane potential of −70 mV with a K-gluconate based internal recording solution, or using a cesium-methanesulfonate internal solution (data not shown). This finding is consistent with morphological evidence of larger cell somata (*Figure 6A*). The inputs to these larger cells also appear to be abnormal. Dlx-CKO LCIs displayed significantly more spontaneous inhibitory postsynaptic currents (sIPSCs) than control LCIs (*Figure 7D,E*) (p = 0.006). Although the mean spontaneous excitatory postsynaptic current (sEPSC) frequency also was greater for Dlx-CKO LCIs (*Figure 7D,F*) the difference from the control LCI mean was not statistically significant (p = 0.17). The sIPSC/sEPSC ratio was significantly higher in Dlx-CKO cells (*Figure 7G*) indicating that these cells receive abnormal synaptic input.

Several abnormalities were also identified in the response of LCIs to various stimuli. Depolarizing current pulses (1 s duration, 25 pA increments) produced significantly fewer action potentials in LCIs from Dlx-CKO than control at 150–200 pA current intensities (*Figure 7H,I*; two-way ANOVA, interaction between current intensity and frequency of action potentials, p < 0.001). These findings indicate that Dlx-CKO LCIs may be less excitable than LCIs from control mice. However, evoked EPSCs from

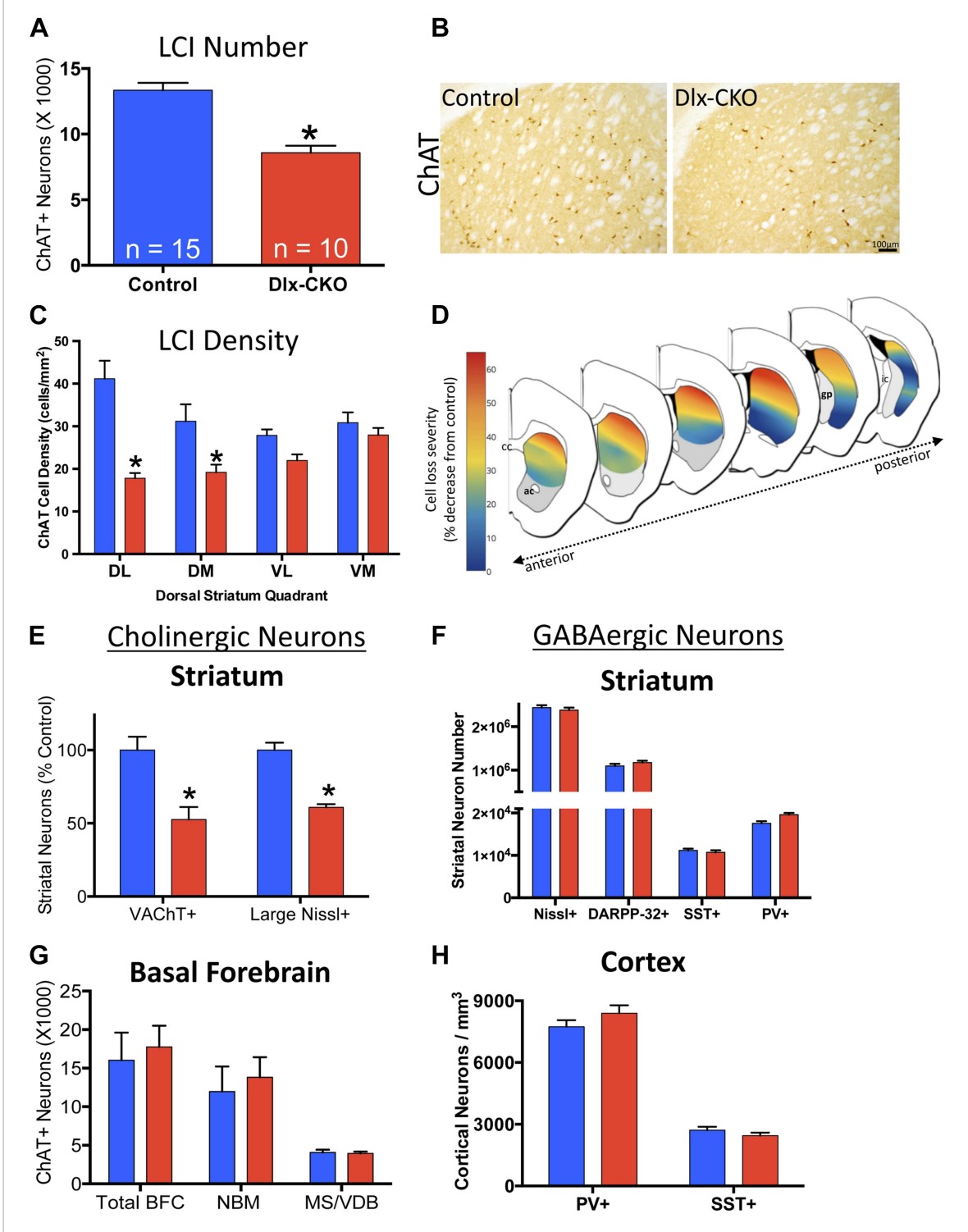

**Figure 4.** Large cholinergic interneurons are selectively Lost from the striatum of Dlx-CKO mice. (**A**, **B**) Stereological quantification of the number of ChAT-positive neurons in the striata of Dlx-CKO and littermate control mice (t-test: $t_{(23)}$ = 5.87; $p < 0.0001$). (**C**) Characterization of the topology of ChAT-positive cell loss in dorsal striatum. Significant decreases in ChAT-positive cells were observed only in the dorsal quadrants. Two-way ANOVA main effects of genotype: $F_{1,56}$ = 38.17; $p < 0.0001$ and interaction: $F_{3,56}$ = 6.405; $p$ = 0.0008. (**D**) Pseudocolor representation of the degree of ChAT-positive cell loss in the dorsal striatum of Dlx-CKO mice. (**E**) Stereological quantification of the number of VAChT-positive and large ($\geq$20 μm diameter soma) Nissl-stained cells. VAChT $t_{(13)}$ = 3.305; $p$ = 0.005, Nissl $t_{(13)}$ = 5.293; $p$ = 0.0001. (**F**) Stereological quantification of the number of striatal small/medium (<20 μm diameter soma) nissl-positive cells (nissl+, $t_{(13)}$ = 0.606; $p$ = 0.549), medium spiny neurons (DARPP-32+: $t_{(22)}$ = 1.14; $p$ = 0.266), and SST- and
*Figure 4. continued on next page*

*Figure 4. Continued*

PV-expressing inhibitory interneuron classes (PV+: $t_{(23)}$ = 2.806, p = 0.01 SST+: $t_{(23)}$ = 0.6865; p = 0.499). (**G**) Stereological quantification of the number of ChAT-positive neurons in basal forebrain nuclei (BFC—Basal Forebrain Complex, MS—Medial Septum, VDB—Vertical Limb of the Diagonal Band) of Dlx-CKO and littermate control mice ($t_{(7)}$ = 0.392; p = 0.706). (**H**) Stereological quantification of the number of cortical SST- and PV-expressing inhibitory interneuron classes (PV+: $t_{(15)}$ = 1.32; p = 0.206; SST+: $t_{(15)}$ = 1.18; p = 0.256).

The following figure supplements are available for figure 4:

**Figure supplement 1**. LCI cell loss is most prominent in dorsolateral striatum.

**Figure supplement 2**. TorsinA is deleted from basal forebrain cholinergic neurons in Dlx-CKO mice.

Dlx-CKO LCIs were significantly larger than those of control LCIs (*Figure 7J,K*) at higher stimulation intensities (two-way ANOVA, interaction between current intensity and response amplitude, p < 0.01). A subpopulation of Dlx-CKO LCIs displayed very large responses (>150 pA at 0.06 mA, 3/12) that were not observed in control LCIs. LCIs exhibiting large amplitude responses also had significantly larger membrane capacitances than cells that did not (134.8 ± 14.7 vs 97.0 ± 8.3 pF, p = 0.047), suggesting that increased numbers of synapses on the larger somata may account for the increased response amplitude. In support of this possibility, current density measurements of evoked responses (evoked response amplitude divided by cell capacitance) were not significantly different at all stimulation intensities (data not shown). These multiple disturbances of surviving LCIs function raise the possibility that the beneficial effect of anticholinergic agents may in part arise from suppressing their aberrant signaling.

## DYT1 postmortem putamen shows a selective reduction of cholinergic markers

Our results suggest that the selective loss of LCIs may be a pathogenic event in DYT1 dystonia. We further explored this possibility by analyzing postmortem putamen from DYT1 subjects and controls. As DYT1 tissue is in very limited supply and in general not of sufficient quality to perform valid quantification of cell numbers, we analyzed whole cell lysates of this tissue for cholinergic (TrkA, VAChT and AChE), GABAergic (GAD67), and dopaminergic (TH) markers (*Figure 8A*; *Table 1*). TrkA levels were significantly reduced in DYT1 putamen (*Figure 8B*; approximately 84% reduction). Similarly, the normalized mean expression levels of VAChT and AChE were reduced by 66% and 50% respectively (*Figure 8B,C*), but these differences did not reach statistical significance, likely because of the large variability between control subjects for these markers. There was also considerable variability in the expression levels of the biosynthetic enzymes for ACh, GABA, and dopamine. Mean ChAT expression levels were slightly higher in DYT1 patients as compared to control subjects (*Figure 8A,D*). GAD67 and TH levels were comparable between control and DYT1 dystonia subjects (*Figure 8E,F*). Consistent with our experimental studies, these results suggest dysregulation of cholinergic function and TrkA signaling in the putamen of DYT1 dystonia patients.

## Discussion

Our studies establish the first model of DYT1 dystonia that exhibits face, construct, and predictive validity. This model incorporates several features believed essential for disease pathogenesis, including torsinA loss of function and targeting of forebrain motor circuitry including GABAergic and cholinergic neurons. Similar to the natural history and therapeutic response of the human disease, Dlx-CKO mice develop overt, dystonic-like twisting movements during juvenile CNS maturation that are reduced significantly with antimuscarinic drugs. The onset of abnormal movements coincides with a selective degeneration of striatal LCIs. Dlx-CKO mice are one of a small number of DYT1 mutant mouse models that exhibit overtly abnormal twisting movements (*Liang et al., 2014*), all of which exhibit loss of discrete populations of neurons critical for motor function. Clasping and twisting during tail suspension is observed in models of many neurological diseases, so are unlikely to be equivalent to dystonia. Nevertheless, this abnormal behavior is an overt manifestation of abnormal motor function linked to cell loss in Dlx-CKO and other symptomatic dystonia models (*Liang et al., 2014*).

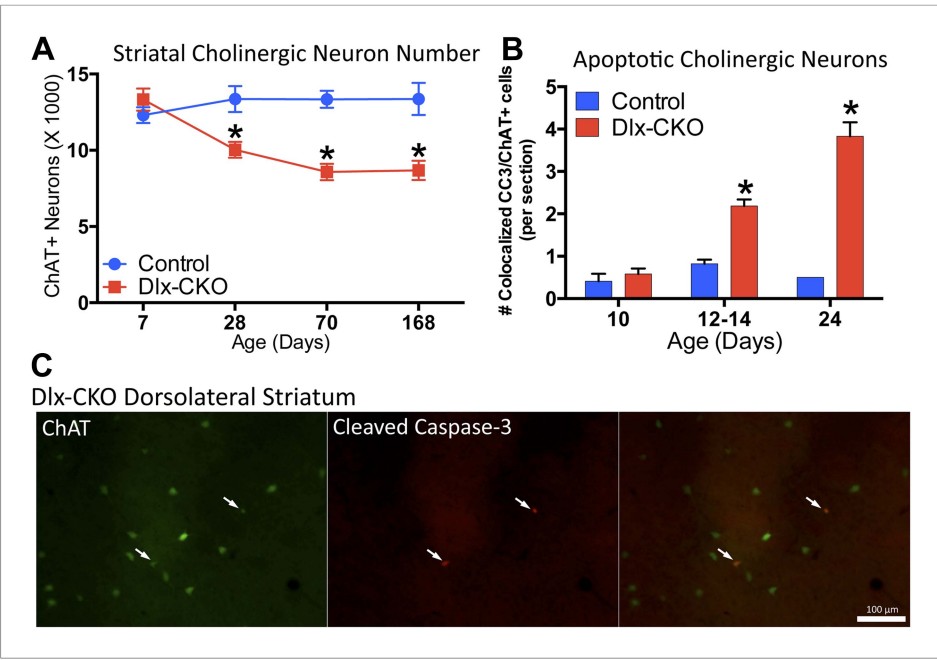

**Figure 5**. Dlx-CKO LCIs degenerate during juvenile striatal maturation, coincident with the onset of abnormal twisting. (**A**) Stereological quantification of the number of ChAT-positive neurons in the striata of Dlx-CKO and littermate control mice at time points between postnatal day 7 and 168. Two-way ANOVA main effects of age: $F_{3,66} = 2.899$; p = 0.04, genotype: $F_{1,66} = 33.74$; p < 0.0001, and interaction: $F_{3,66} = 7.232$; p = 0.0003; * represents time points where significant differences exist using Sidak's multiple comparison test. (**B**, **C**) Quantification of the number of ChAT-positive striatal neurons co-expressing cleaved caspase-3 between P10 and P24 in control and Dlx-CKO brain sections (two-way ANOVA main effects of age $F_{2,21} = 43.68$; p < 0.0001, genotype: $F_{1,21} = 122.1$; p < 0.0001, and interaction $F_{2,21} = 32.91$; p < 0.0001).

The following figure supplements are available for figure 5:

**Figure supplement 1**. ChAT and TrkA expression is normal at P7.

**Figure supplement 2**. There are no differences in the number of non-cholinergic apoptotic striatal cells.

---

This is the first demonstration of selective vulnerability of a striatal cell type to torsinA loss of function, findings supported by studies in postmortem putamen from DYT1 dystonia subjects. Our observations add to an emerging literature implicating CNS maturation-related vulnerability of discrete cell types in the pathogenesis of DYT1 dystonia, a disease mechanism with potentially important implications for future approaches to therapy.

The onset and progression of Dlx-CKO motor abnormalities as juveniles, followed by a fixed defect that persists for life (*Figure 1*), broadly resembles the progression of DYT1 and other childhood-onset primary dystonias. Abnormal behavior onset and striatal LCI loss in Dlx-CKO mice coincides with the period of striatal circuit maturation when connections are established and projection- and interneurons become physiologically active and mature (*Hattori and McGeer, 1973*; *Tepper et al., 1998*; *Lee and Sawatari, 2011*). Subjects that carry the DYT1 mutation but do not manifest dystonia by their mid-20's almost invariably remain asymptomatic for life (*Bressman et al., 2000*). These clinical and experimental observations support a critical role for torsinA during CNS maturation, and suggest that torsinA deficiency during a defined developmental window is a key component of DYT1 pathogenesis.

Previous torsinA loss-of-function models also demonstrate cell loss restricted to a neurodevelopmental window (*Liang et al., 2014*). These observations suggest that CNS maturation, synaptogenesis, and associated processes may exert a unique stress on neuronal circuits. TorsinA is a AAA+ protein chaperone (*Ozelius et al., 1997*) with putative functions in protein quality control (*Chen et al., 2010*; *Nery et al., 2011*) regulation of nucleocytoskeletal connections (*Gerace, 2004*; *Worman and Gundersen, 2006*), trafficking of membrane-bound proteins (*Torres et al., 2004*), protein secretion

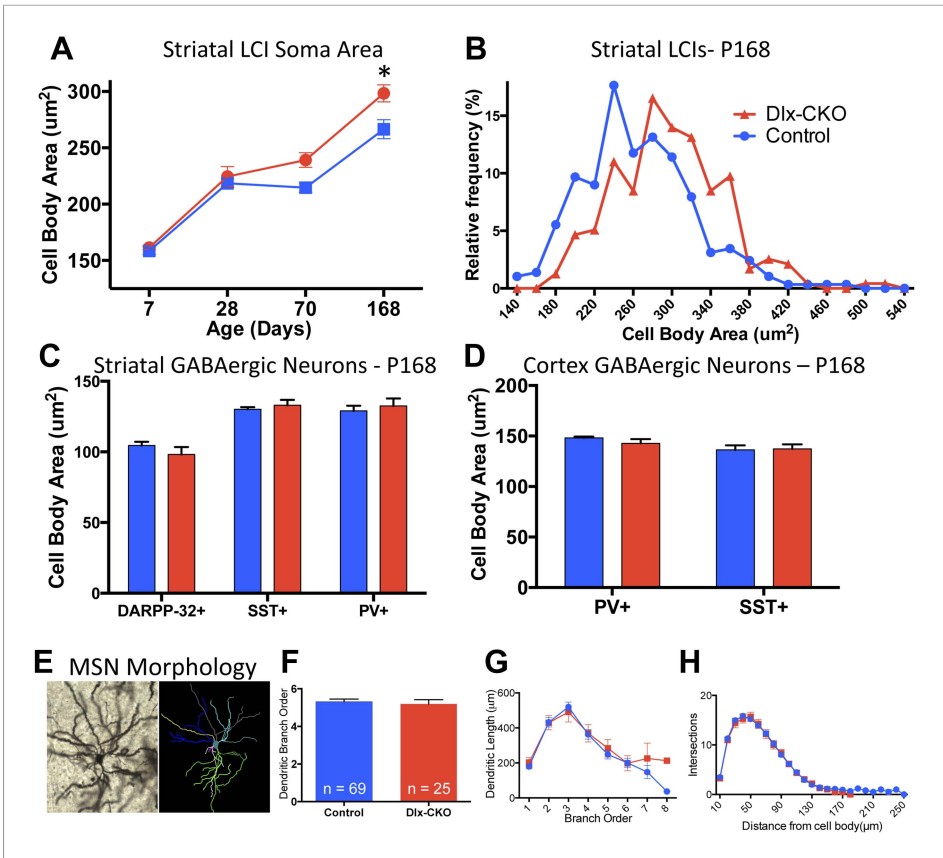

**Figure 6**. Surviving striatal LCIs exhibit cell soma hypertrophy. (**A**) Quantification of ChAT-positive cell soma area in dorsal striatum between postnatal day 7 and 168 (two-way ANOVA significant main effect of genotype $F_{1,60} = 12.51$; $p = 0.0008$ and time $F_{3,60} = 117.8$; $p < 0.0001$, Tukey's multiple comparison test). (**B**) Frequency histogram of cell soma area data at postnatal day 168. (**C**, **D**) Cell soma area of striatal and cortical GABAergic interneurons and striatal MSNs at postnatal day 168. (**E**) Example of Golgi-Cox-stained MSN and dendritic tree reconstruction. (**F**–**H**) Analysis of dendritic complexity (n = 69 control, 25 Dlx-CKO neurons). No differences observed in average highest dendritic branch order (one-way ANOVA $F_{3,90} = 1.079$; $p = 0.36$), dendritic length (one-way ANOVA $F_{3,92} = 1.023$; $p = 0.386$), or intersections on sholl analysis (two-way ANOVA $F_{1,92} = 0.019$; $p = 0.89$).

(*Hewett et al., 2007, 2008*), and nuclear export of large ribonucleoprotein granules during synapse development (*Jokhi et al., 2013*). Periods of synaptogenesis and circuit maturation are therefore periods in which torsinA function may be particularly important. Cell loss and behavior onset during a neurodevelopmental window is consistent with a critical period for torsinA function in CNS development.

LCIs play an early role in striatal development, which may in part account for the differences observed between Dlx-CKO mice and previous models. LCIs are the first striatal neurons to become post-mitotic (*Phelps et al., 1989*), and cholinergic signaling is believed to have important but poorly understood influences on the development of later-born striatal neurons (*Chesselet et al., 2007*). In contrast to our findings using the *Dlx5/6-Cre* transgene, which is active prior to these events (in progenitor cells), *postnatal* conditional deletion of *Tor1a* in cholinergic neurons does not cause overt abnormal movements or changes in striatal LCI number, morphology, and neurochemistry (*Sciamanna et al., 2012a*). Consistent with an important role for timing of deletion, *germline* heterozygous ΔE *Tor1a* knock-in mice exhibit increased LCI cell body area, though these mutants show no difference in the number of LCIs (*Song et al., 2013*). Mice may require more marked torsinA loss of function to fully model cholinergic vulnerability, as the heterozygous ΔE *Tor1a* knock-in mice retain considerable torsinA function, and do not exhibit overt abnormal movements (*Tanabe et al., 2012*). It is possible that non-cholinergic neuron populations in Dlx-CKO mice appear morphologically normal but exhibit functional abnormalities. However, conditional

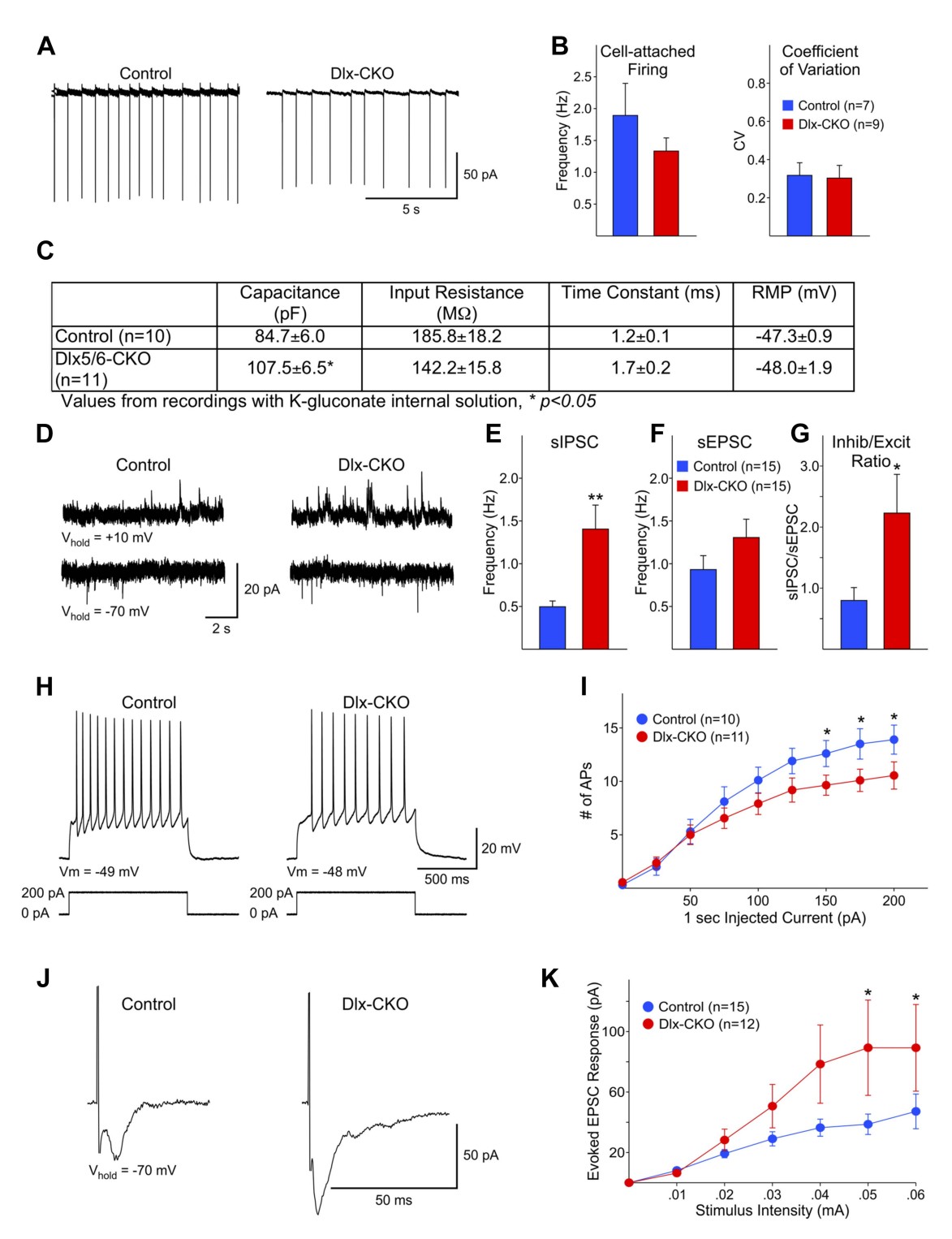

**Figure 7**. Surviving LCIs exhibit changes in excitability and abnormal synaptic inputs. (**A**) Sample cell-attached recordings from tonically active control and Dlx-CKO LCIs. (**B**) Mean frequencies of spontaneous firing (cell-attached) and coefficients of variation from control and Dlx-CKO LCIs. (**C**) Capacitance, input resistance, time constant, and resting membrane potential values from recordings with K-gluconate internal solution. (**D**) Sample recordings of sIPSCs and sEPSCs. (**E**) Mean sIPSC frequency from Dlx-CKO LCIs was significantly greater than that of control LCIs (p = 0.006). (**F**) Mean sEPSC frequencies from both genotypes were similar. (**G**) Ratio of sIPSC/sEPSC indicates that Dlx-CKO LCIs received significantly more inhibitory inputs than control LCIs (p = 0.05). (**H**) Examples of typical responses of control and Dlx-CKO LCIs to injected current pulses. Control LCIs generated more action
*Figure 7. continued on next page*

**Figure 7. Continued**

potentials. (**I**) Mean numbers of action potentials are significantly reduced in Dlx-CKO LCIs at higher injected currents (two-way ANOVA with posthoc Bonferroni test, p < 0.001). (**J**) Sample traces of evoked EPSCs in control and Dlx-CKO LCIs. (**K**) Peak amplitudes of evoked EPSCs were significantly larger in Dlx-CKO LCIs (two-way ANOVA with posthoc Bonferroni test, p < 0.01).

deletion of torsinA in MSNs, which constitute 95% of all striatal neurons, does not cause overt abnormal movements (*Yokoi et al., 2011*).

Current dogma holds that neurodegeneration does not occur in primary dystonia, but postmortem samples from DYT1 subjects are limited and have not been assessed quantitatively for abnormalities in the number of striatal neuronal cell types (*Paudel et al., 2012*). Neuroimaging studies increasingly point to changes potentially consistent with cell loss (reviewed in *Ramdhani and Simonyan (2013)*). Our studies in postmortem striatum from DYT1 subjects, the first to examine cell type-specific markers in this context, are consistent with our experimental findings pointing to selective cholinergic defect in this structure. Previous work in overtly symptomatic DYT1 models demonstrates loss of discrete cell types in other motor-related structures (e.g., loss of neurons of the deep cerebellar nuclei, with preservation of all other cerebellar cell types (*Liang et al., 2014*). Our analysis of postmortem tissue was restricted to the striatum; future work is required to determine whether there is cell loss in extra-striatal areas corresponding to those identified in murine models. Postmortem analyses of DYT1 subjects suggests the presence of inclusion bodies in pedunculopontine cholinergic neurons, but did not report cell loss (*McNaught et al., 2004*). These observations, and additional work (*Liang et al., 2014*) and reviewed by *Dauer (2014)*, suggest that abnormal protein quality control may represent a pathogenic event in DYT1 dystonia.

Several lines of evidence are consistent with a potentially causal connection between LCI loss or dysfunction and abnormal twisting movements. Unilateral immunotoxin-mediated striatal cholinergic ablation causes 'asymmetric concaving postures' in mice (*Kaneko et al., 2000*) and LCI ablation also reduces the threshold for the development of motor tics (*Xu et al., 2015*). LCIs are lost in a perinatal hypoxia-ischemia model of cerebral palsy-associated dystonia, mimicking the striatal pathology seen in that disease ('status marmoratus') (*Burke and Karanas, 1990*), which is also treated with anticholinergic drugs. Furthermore, imaging studies of idiopathic cervical dystonia subjects suggest deficits in the density of striatal cholinergic axon terminals (*Albin et al., 2003*). LCIs innervate fast-spiking interneurons (*Koos and Tepper, 2002*), a cell type concentrated in the lateral striatal areas (*Berke, 2011*) where LCI loss is greatest in Dlx-CKO mice. Fast-spiking interneurons contribute to action execution and suppression of unwanted movements (*Gage et al., 2010*), and have been implicated in dystonic-like postures (*Gittis et al., 2011*).

Previous reports of DYT1 rodent models support a role between torsinA loss of function and aberrant physiological properties of LCIs (*Eskow Jaunarajs et al., 2015*), but these models do not exhibit abnormal twisting movements or cell loss (*Sharma et al., 2005*; *Pisani et al., 2006*; *Martella et al., 2009*; *Quartarone and Pisani, 2011*; *Sciamanna et al., 2011*, *2012b*). Abnormal clasping is present in a transgenic rat model of dystonia, but it is not known if striatal LCI number is reduced in these animals (*Grundmann et al., 2012*). Loss of cholinergic neurons may be required for the remaining cells to become interconnected into an abnormal circuit. Recordings in striatal slices performed after maximal cell loss in Dlx-CKO mice demonstrate that surviving LCIs develop a complex set of intrinsic membrane and synaptic alterations. Dlx-CKO LCIs are less excitable when depolarized by intracellular current injections, yet display significantly larger evoked EPSCs, potentially reflecting increased excitatory synapses on the surviving (larger) neurons. Dlx-CKO LCIs also receive significantly enhanced inhibitory synaptic that could lead to aberrant firing when cells are challenged, as we have demonstrated in a related movement disorder, Huntington's chorea (*Holley et al., 2015*). These multiple disturbances of surviving LCIs function raise the possibility that the beneficial effect of anticholinergic agents may in part arise from suppressing aberrant LCI signaling.

Our findings add to the emerging literature demonstrating a unique requirement for torsinA function for cell viability during CNS maturation (*Liang et al., 2014*), a concept that may alter approaches to the therapeutic targeting of DYT1 dystonia. These experiments also raise the possibility that unique classes of striatal LCIs may exist that vary in their molecular and functional features. Future studies aimed at identifying factors that modulate the vulnerability of these cells to torsinA loss-of-function may help to define such classes, advancing our understanding of striatal organization and function.

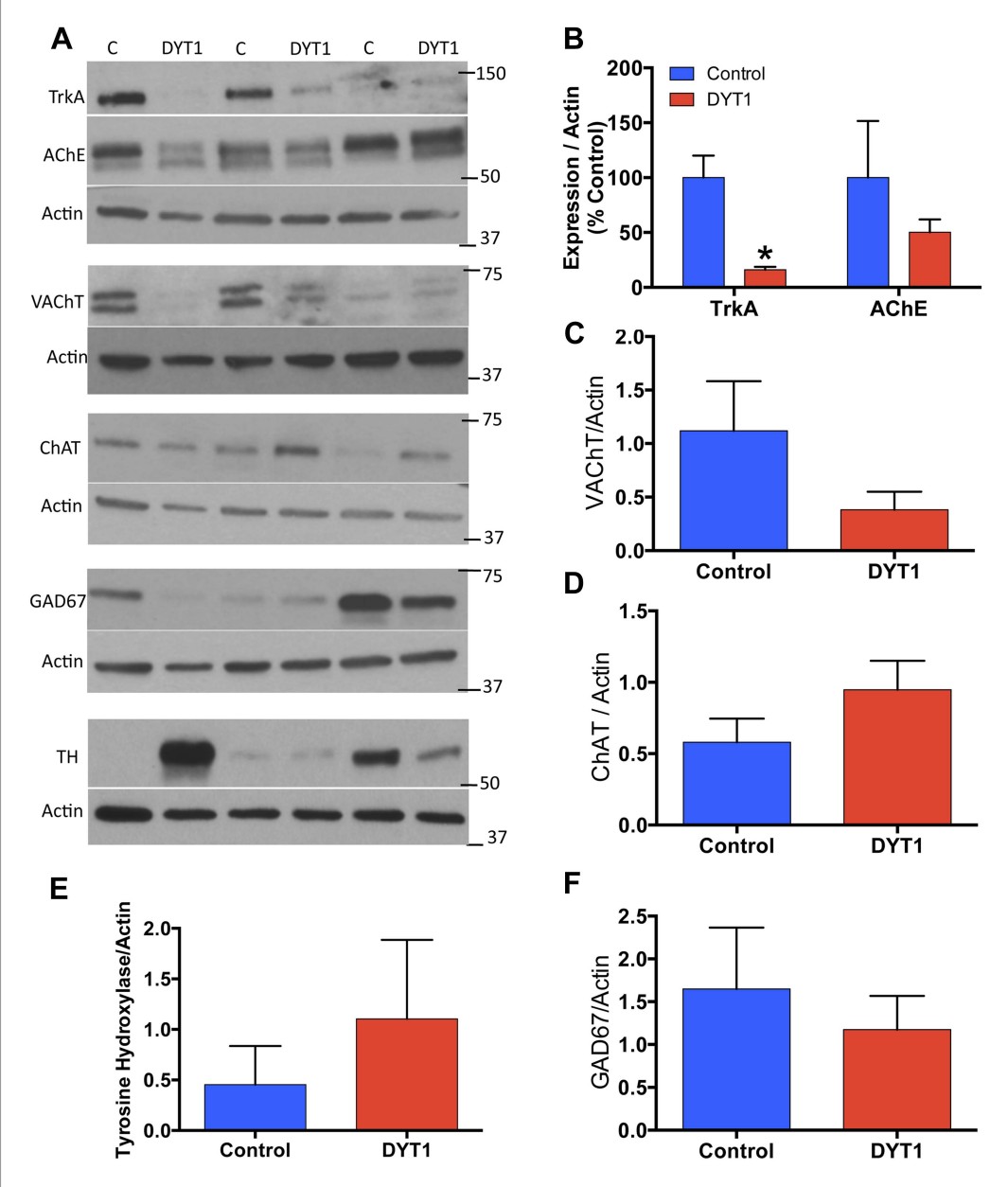

**Figure 8**. DYT1 dystonia postmortem putamen displays selective reductions in cholinergic markers. (**A**) Western Blot analysis of postmortem putamen samples from 3 dystonia patients and 3 age-matched control subjects. (**B**) Significant reductions in TrkA expression (t-test; $t_{(4)}$ = 4.413; p = 0.014). (**B–F**) No significant alterations in AChE ($t_{(4)}$ = 0.940; p = 0.400), VAChT ($t_{(4)}$ = 0.208; p = 0.208), ChAT ($t_{(df = 4)}$ = 1.766; p = 0.152), TH ($t_{(4)}$ = 0.7459; p = 0.497), or GAD67 ($t_{(4)}$ = 0580; p = 0.593).

# Materials and methods

## Generation and maintenance of mice

*Tor1a* floxed mice were generated as previously described (*Liang et al., 2014*). *Dlx5/6-Cre* mice were obtained from Jackson laboratories (Tg(*dlx6a-Cre*)1Mekk/J; stock number 008199) and then maintained in our mouse colony at the University of Michigan. Conditional *Tor1a* null animals were generated with the following breeding scheme: *Cre$^+$ Tor1a$^{+/−}$* X *Tor1a$^{flx/flx}$*, with four possible offspring genotypes: *Tor1a$^{flx/+}$* (WT), *Tor1a$^{flx/−}$* (Flx control), *Cre$^+$ Tor1a$^{flx/+}$* (Cre control), and

**Table 1**. Human subject data

| Case I.D. number | Lane # | Status | Age (years) | Cause of death | Other neuro-pathology | Sex | PMI (hrs) | Time in storage | Race |
|---|---|---|---|---|---|---|---|---|---|
| BBID100 | 1 | Control | 87 | Unknown | – | Female | 9 | 10 year 2 month | Caucasian |
| UMB1619 | 2 | DYT1 | 87.8 | Stroke | – | Female | 23 | 12 year 4 month | Caucasian |
| BBID384 | 3 | Control | 89 | Respiratory failure | – | Female | Not recorded | 11 year 9 month | Caucasian |
| UMB4877 | 4 | DYT1 | 90.3 | Stroke | – | Female | 2 | 6 year 6 month | Caucasian |
| BBID732 | 5 | Control | 91 | Unknown | Lacunar infarctions, cerebellar microinfarctions, modest nigral cell loss, lewy bodies | Male | 8 | 10 year 10 month | Caucasian |
| UMB5200 | 6 | DYT1 | 88.8 | 'complications of disorder' | – | Female | 9 | 5 year 5 month | Caucasian |

*Cre$^+$ Tor1a$^{flx/-}$* (Dlx-CKO). Mice were genotyped for *Tor1a* and the *Cre* transgene using the primers and PCR programs previously described (*Liang et al., 2014*). For electrophysiology experiments, a *Chat*(BAC)-eGFP allele (strain B6.Cg-Tg(RP23-268L19-EGFP)2Mik/J; Stock Number 007902) was bred into the Dlx-CKO cross to allow for visualization of LCIs. Mice were housed 2–4 per cage using microisolation technique, maintained in a temperature- and light-controlled room, and provided with food and water ad libitum. Mice of all genotypes were housed together to prevent environmental bias. The University of Michigan Committee on the Use and Care of Animals (UCUCA) approved all experiments involving animals. Age and sex-matched littermate mice were used for all experiments.

## Behavioral analysis

### Pre-weaning behavioral observation
Mice were examined for motor reflexes between postnatal days 1 and 21. Surface righting reflex, postural reflex, negative geotaxis, and forelimb wire suspension were performed and quantified as previously described (*Santos et al., 2007*).

### Tail suspension test
Mice were picked up by the tail, suspended in the air for 60 s, and were observed for forelimb and hindlimb clasping and trunk twisting. Mice were periodically tested for the presence of abnormal clasping and twisting behaviors between postnatal day 7 and 1 year of age. 2–3 investigators blinded to genotype rated the presence of abnormal clasping or twisting for each mouse cohort.

### Open field
8-week-old mice were placed in one of four 43 × 43 cm$^2$ square plexiglass cages equipped with infrared beams to detect horizontal movements and vertical rearing (MED Associates, St. Albans, VT) and were monitored for 60 min. All horizontal and vertical beam breaks were recorded. Data expressed as number of beam breaks per 5-min epoch.

### Grid hang test
Mice were tested for motor coordination and strength by placing them on a 21.5 × 21.5 cm wire grid (0.5 cm$^2$ openings) and turning upside down 21 cm above the bench. Latency to fall was recorded, with a cut off time of 300 s. Mice were observed throughout the test as they readjusted grip and made new paw placements.

### Accelerating rotarod
Motor coordination, endurance, and motor learning were assessed by placing mice on a stationary rod, and increasing speed of rotation from 4 to 40 rpm over the course of 5 min. Latency to fall was recorded, with a cut-off time of 5 min. Mice were tested in 5 consecutive trials per day for 2 days, with 1-min rest between trials.

## Gait analysis

8-week old mice were tested using a Digigait apparatus (Mouse Specifics Inc., Framingham, MA) with a treadmill speed of 24 cm/s. Dynamic gait signals were generated with digigait software, an investigator blinded to genotype and experimental design confirmed the accuracy of digital paw prints, and 10 gait indices were examined.

## Response to anticholinergics

Repeated daily tail suspension tests were administered beginning at 10 weeks of age. Once-daily injections of scopolamine (3 mg/kg, s.c.; Sigma S0929) or saline (10 ml/kg) were administered for 10 days, and mice received 1-minute tail suspensions 45 min after each injection. A separate cohort of mice received once-daily injections of trihexyphenidyl (THP; 5 mg/kg, i.p.; Sigma T1516) or saline (10 ml/kg) for 5 days and was tested with 1-minute tail suspensions 45 min after each injection. Mice were tested again 3 weeks after the last injection for both experiments. Each tail suspension was recorded and 2 (for scopolamine study) or 3 (for THP study) observers blinded to experimental design, groups, and trial rated clasping behaviors. The presence of forelimb or hindlimb clasping and trunk twisting was timed, with a maximum value of 60 s.

## Western blot analysis

Mice were sacrificed with cervical dislocation, brains were rapidly removed, frozen over dry ice, and stored at −80°C. Fresh frozen brains were cut into 500 μm sections, were refrozen onto uncoated glass slides, and striatum samples were taken using a modified 16 gauge needle. Micropunch samples were placed into microcentrifuge tubes containing 100 μl lysis buffer (Tris Buffered Saline containing 1% Sodium Dodecyl Sulfate, 0.1 mM phenylmethanesulfonyl fluoride, 1 mM Dithiothreitol, and Halt Protease Inhibitor Cocktail [Life Technologies product 87786]) and were homogenized using a plastic plunger. Homogenates were centrifuged at 12,000 rpm for 5 min, pellets were discarded, and the supernatants were removed to a new tube. Bradford protein assay was performed and final lysates were prepared at 1 μg/μl, including sample-loading buffer (0.05% bromophenol blue, 0.1 M dithiothreitol, 10% glycerol, 2% SDS, and 5% β-mercaptoethanol), and were boiled for 5 min. 10 μg (10 μl) protein samples and Dual Precision Plus protein standards were run on 4–15% Biorad Mini Protean TGX precast polyacrylamide gels, underwent wet transfer to 0.22 μm PVDF membranes in transfer buffer containing 10% methanol (run for 2 hr at 400 mA at 4°C), and were processed for enhanced chemiluminescence as described below. Membranes were washed in tris buffered saline (TBS) containing 1% Tween-20 (TBS-T), blocked for 30 min in 5% non-fat dry milk in TBS-T, and incubated in primary antibody overnight at 4°C (see *Table 2* for primary antibody details). Membranes were then washed in 5% milk/TBS-T, incubated for 1 hr in horseradish peroxidase-conjugated secondary antibody (*Table 2*), and rinsed in TBS-T. Bands were visualized using Supersignal West Pico, Dura, or Femto enhanced chemiluminescence substrates, underwent multiple exposures to Amersham hyperfilm ECL, and were developed and fixed with an x-ray film developer. Films were scanned using an Epson scanner, and band intensity was quantified in ImageJ. Serial dilutions of protein were examined for each antibody to determine optimal antibody dilutions before running experimental samples, and multiple exposures were examined to confirm that bands were in the linear range and not overexposed.

## In vivo microdialysis

Custom made microdialysis probes with 1 mm polyacrylonitrile membrane length were implanted bilaterally into the dorsal striatum 24 hr before experiments with the following coordinates: Anteroposterior +1.1 mm, mediolateral +2.05 mm, dorsoventral −3.8 mm. On the day of experiments, artificial cerebrospinal fluid (aCSF) (composition in mM: $CaCl_2$ 1.2; KCl 2.7, NaCl 148 and $MgCl_2$ 0.85) was perfused through the microdialysis probe at 2 μl/min for 1 hr and then 1 μl/min for another hour for equilibration. For studies that only measured ACh concentrations (*Figure 3E*), three 5-min fractions were collected per animal following neostigmine addition (50 μM) to the aCSF perfusate. Dialysate samples were collected and analyzed following the addition of d4-ACh (20 nM) as internal standard. For comprehensive neurochemical analysis of basal differences between genotypes (*Figure 3F*), five 3-min fractions were collected per animal, and a benzoyl chloride derivatization scheme was employed (*Song et al., 2012*). Briefly, 2.5 μl borate buffer (100 mM), 2.5 μl 2% benzoyl chloride in acetonitrile, and 2.5 μl internal standard solution was added to each dialysate sample prior to analysis.

**Table 2**. Antibodies used for immunohistochemistry and western blots

|  | Level | Antigen | Host | Conjugated | Dilution | Source |
|---|---|---|---|---|---|---|
| IHC | Primary | TorsinA | Rabbit | – | 1:100 | Abcam ab34540 |
|  | Primary | GFAP | Rabbit | – | 1:2000 | Dako Z0334 |
|  | Primary | s100β | Rabbit | – | 1:2000 | Abcam ab41548 |
|  | Primary | Iba-1 | Rabbit | – | 1:500 | Wako 019-19741 |
|  | Primary | ChAT | Goat | – | 1:100 | Millipore AB144P |
|  | Primary | VAChT | Goat | – | 1:2000 | Millipore ABN100 |
|  | Primary | DARPP-32 | Rabbit | – | 1:300 | Cell Signaling #2302 |
|  | Primary | PV | Mouse | – | 1:500 | Swant #235 |
|  | Primary | SST | Rabbit | – | 1:500 | Abcam ab103790 |
|  | Primary | CC3 | Rabbit | – | 1:500 | Cell Signaling #9664 |
|  | Secondary | anti-mouse | Donkey | Ax488 | 1:800 | Life Technologies A-31572 |
|  | Secondary | anti-mouse | Donkey | Ax555 | 1:800 | Life Technologies A-21202 |
|  | Secondary | anti-mouse | Donkey | biotin | 1:800 | Jackson Immunoresearch 115-065-003 |
|  | Secondary | anti-rabbit | Donkey | Ax488 | 1:800 | Life Technologies A-21206 |
|  | Secondary | anti-rabbit | Donkey | Ax555 | 1:800 | Life Technologies A-31572 |
|  | Secondary | anti-rabbit | Donkey | biotin | 1:800 | Jackson Immunoresearch 711-065-152 |
|  | Secondary | anti-goat | Donkey | biotin | 1:800 | Jackson Immunoresearch 705-065-003 |
| Western blot | Primary | ChAT | Rabbit | – | 1:1000 | Abcam ab137349 |
|  | Primary | GAD67 | Mouse | – | 1:1000 | Millipore MAB5406 |
|  | Primary | TH | Rabbit | – | 1:2000 | Millipore AB152 |
|  | Primary | Actin | Mouse | – | 1:6000 | Sigma A5316 |
|  | Primary | TrkA | Rabbit | – | 1:4000 | Advanced Targeting Systems ABN03 |
|  | Primary | DARPP-32 | Rabbit | – | 1:2000 | Cell Signaling #2302 |
|  | Primary | TorsinA | Rabbit | – | 1:10,000 | Abcam ab34540 |
|  | Primary | Calnexin | Rabbit | – | 1:20,000 | Enzo Life Sciences SPA-860 |
|  | Primary | AChE | Rabbit | – | 1:200 | Santa Cruz sc-11409 |
|  | Primary | VAChT | Goat | – | 1:1000 | Millipore ABN100 |
|  | Primary | TrkA (for human) | Rabbit | – | 1:1000 | Cell Signaling #2505 |
|  | Secondary | Anti-goat | Rabbit | HRP | 1:7500 | Pierce 31402 |
|  | Secondary | anti-mouse | Goat | HRP | 1:5000 | Jackson Immunoresearch 115-035-003 |
|  | Secondary | anti-rabbit | Goat | HRP | 1:10,000 | Jackson Immunoresearch 111-035-003 |

## HPLC-MS analyses of neurochemistry

Following sample collection, a Thermo Scientific Accela HPLC (Waltham, MA) system automatically injected 5 μl of the sample onto a Waters (Milford, MA) HSS T3 reverse phase HPLC column (1 mm × 100 mm, 1.8 μm) at 200 μl/min. For ACh analysis, a 2 min isocratic elution was employed (25/75 mobile phase A/B). Mobile phase A consisted of 10 mM ammonium formate and 0.15% formic acid. Mobile phase B was acetonitrile. Analytes were detected by a Thermo Scientific TSQ Quantum Ultra triple quadrupole mass spectrometer operating in multiple reaction monitoring mode. ACh was detected using the m/z transition 146‡87 while d4-ACh was detected using 151‡90. For comprehensive neurochemical analysis, samples were analyzed as previously described but with a 6 min HPLC gradient (*Song et al., 2012*).

## Histology and immunohistochemistry

Mice were deeply anesthetized with a lethal dose of ketamine/xylazine and received transcardial perfusion of 0.01 M phosphate buffered saline (PBS) followed by 4% paraformaldehyde in 0.1 M phosphate buffer

(PB). Brains were postfixed in 4% paraformaldehyde for 2 hr and cryoprotected overnight in 20% sucrose in PB. Consecutive serial 40 μm brain sections through the forebrain were generated on a cryostat and stored in PBS. Free-floating brain sections were processed for fluorescence immunohistochemistry by washing in PBS containing 0.1% Triton-X-100 (PBS-Tx), blocking in 5% normal donkey serum (NDS), and incubating in primary antibody overnight at 4°C (see *Table 2* for primary antibody details). Sections were washed in PBS-Tx followed by 1 hr in secondary antibodies conjugated to Alexafluor 488 or Alexafluor 555 (*Table 2*). Brain sections were mounted onto gelatin-coated slides, coverslipped with prolong gold antifade mounting medium, and imaged under epifluorescence microscopy. Free-floating brain sections were processed for DAB staining by washing in PBS-Tx, blocking with 0.3% $H_2O_2$ followed by 5% NDS, and incubating overnight in primary antibody overnight at 4°C (see *Table 2*). Sections were washed with PBS-Tx, incubated in biotinylated secondary antibody for 1 hour (*Table 2*), followed by 2 hr in avidin-biotin-peroxidase complex (Vectastain Elite ABC Kit Standard; PK6100, Vector Laboratories, Burlingame, CA). Sections were exposed to 3,3′ diaminobenzidine using Sigmafast DAB tablets (Sigma D4418) and were flooded with PBS to halt staining. Sections were mounted onto gelatin-coated slides, dried overnight, dehydrated in ethanol and xylenes, and were coverslipped with permount mounting medium. Staining for each experiment was performed in parallel by an investigator blinded to experimental group. Omitting the primary or secondary antibody from incubation prevented all staining. To process for Nissl staining, 40 μm brain sections were mounted onto gelatin-coated slides, were dried for 24 hr, incubated in successive decreasing concentrations of ethanol, followed by 3 min in 0.005% cresyl violet solution containing acetic acid, were dehydrated in ethanol followed by xylenes, and were coverslipped using permount mounting medium.

## Cholinesterase histochemistry

Mice were anesthetized with ketamine/xylazine and sacrificed by cervical dislocation. Brains were removed, frozen over dry ice, and stored at −80°C.

25 μm fresh frozen brain sections were generated on a cryostat, were adhered to gelatin-coated slides, dried at room temperature, and stored at −80°C. Sections were stained as previously described (*Geneser, 1987*). Sections were dehydrated with ethanol followed by xylenes, and were coverslipped with permount mounting medium. Brain sections were imaged using brightfield microscopy, and optical density of striatal staining was determined using ImageJ software. Values from the anterior commissure white matter tract were used for a background subtraction value. AChE specificity was confirmed by omitting substrate, substituting butyrylcholinesterase substrates, or including the AChE inhibitor neostigmine in the incubation medium (*Figure 3—figure supplement 1*).

## Golgi-Cox staining and Sholl analysis

Mice were anesthetized with ketamine/xylazine and sacrificed with cervical dislocation. Brains were removed and immediately processed using the FD Rapid GolgiStain Kit (FD Neurotechnologies, Columbia, MD). Processed brains were frozen with dry ice-chilled isopentane, placed on dry ice, and 100 μm brain sections were generated on a cryostat. Brain sections were mounted onto gelatin-coated slides, stained according to the FD Rapid GolgiStain Kit, and coverslipped with permount mounting medium. Slides were observed under brightfield microscopy using a Zeiss Axiophot 2 microscope, first using a 5× objective lens. Striatal medium spiny neurons containing full golgi-cox impregnation without breaks along the dendrites, and no obstructions by neighboring cells were then used for analysis. Neurons were observed using a 63× objective lens, were traced, and reconstructed using Neurolucida software (MBF Bioscience, Williston, VT), and dendritic complexity was determined with Sholl analysis. 94 neurons from 25 animals were used for this study.

## Cell counting

### Stereology

Striatal, motor cortex, and basal forebrain neuron subtypes and striatal volume were quantified with an unbiased stereological approach using the optical fractionator probe in Stereoinvestigator (MBF Bioscience, Williston, VT). Consecutive 40 μm serial sections through the forebrain were separated into a series of 6 wells and stained for ChAT, VAChT, DARPP-32, PV, SST, or Nissl as described above. Sections were observed using brightfield microscopy on a Zeiss Axiophot 2 microscope. Regions of interest were first outlined using a 5× objective lens. 8 sections were observed for each marker, with

a section evaluation interval of 6. Cells within the outlined region were counted using a 63× oil immersion objective, with a 12 µm counting depth, and 1 µm guard zones. Counting frame and sampling grid sizes were determined in pilot studies such that the Gunderson coefficient of error was less than 0.1 for each marker in each brain region (see *Table 3* for specific counting frame and grid sizes). The top of each stained cell body was the point of reference. The corpus callosum, anterior commissure, lateral ventricle, and globus pallidus were used as anatomical boundaries for the striatum. Motor cortical counts were normalized to the measured volume due to the lack of clear boundaries for regional outlining in the cortex.

## Cortical thickness

Slides were examined under brightfield microscopy using a Zeiss Axioskop 2 plus microscope under a 10× objective lens. Images were acquired and the motor cortex was observed from a series of 6 Nissl-stained sections per brain (10–15 brains per genotype). Four bilateral measurements from the dorsal boundary of the corpus callosum to the outer edge of cortical layer 1 were taken per brain section, and were averaged to generate mean cortical thickness (in µm) using Cellsens standard (Olympus, Center Valley, PA).

## Cell size

Striatal and cortical cell size measurements were quantified from images taken under a 40× oil immersion objective lens. Cell soma area was measured ($\mu m^2$) by outlining the edge of each cell body profile in darkly stained neurons that were fully in focus. 300–500 neurons per genotype (from 8–9 brains per group) were measured.

## Cell density

8 serial sections through the striatum were quadrisected according to the boundaries of the corpus callosum and anterior commissure (vertically bisected at the midpoint of the corpus callosum, and horizontally bisected half way between the corpus callosum and the anterior commissure). The area of each striatal quadrant was determined and the number of soma profiles of ChAT-stained neurons was counted within each quadrant using Cellsens standard software. Heat maps were generated according to the cell density values, and were overlaid onto line drawings of the striatum to generate *Figure 4D*.

## CC3/ChAT counts

To quantify the number of apoptotic LCIs, a series of 8 striatal sections costained for ChAT/Cleaved Caspase-3 (CC3) were observed under epifluorescence microscopy. The total number of CC3+ and ChAT/CC3+ cells were quantified per section.

## Electrophysiology

### Brain slice preparation

Detailed methods have been published (*Cepeda et al., 2013*). Dlx-CKO and control mice (86–168 days, 14 male and 12 female) were deeply anesthetized with isoflurane and perfused intracardially with an

**Table 3**. Optical fractionator parameters used for stereological cell counting

| Region | Marker | Counting frame (µm) | Grid size (µm) |
|---|---|---|---|
| Striatum | ChAT | 100 × 100 | 250 × 250 |
| | VAChT | 100 × 100 | 250 × 250 |
| | PV | 120 × 120 | 330 × 330 |
| | SST | 120 × 120 | 330 × 330 |
| | Nissl (large) | 100 × 100 | 250 × 250 |
| | Nissl (small) | 20 × 20 | 600 × 600 |
| | DARPP-32 | 20 × 20 | 600 × 600 |
| Basal forebrain | ChAT (NBM) | 75 × 75 | 250 × 250 |
| | ChAT (MS/VDB) | 75 × 75 | 150 × 150 |
| Cortex | PV | 75 × 75 | 330 × 330 |
| | SST | 75 × 75 | 330 × 330 |

ice-cold sucrose slicing solution containing the following (in mM): 87 NaCl, 2.5 KCl, 0.5 CaCl$_2$, 7 MgCl$_2$, 1.25 NaH$_2$PO$_4$, 26 NaHCO$_3$, and 75 sucrose, pH 7.2 (aerated with 95% O$_2$/5% CO$_2$, 290–300 mOsm/l). Mice were then decapitated and the brain rapidly removed and placed in the ice-cold sucrose slicing solution. Coronal slices of the striatum were cut (350 μm) using a vibrating microtome (VT1000S; Leica Microsystems, Germany), transferred to an incubating chamber containing aCSF (130 NaCl, 3 KCl, 1.25 NaH$_2$PO$_4$, 26 NaHCO$_3$, 2 MgCl$_2$, 2 CaCl$_2$, and 10 glucose) oxygenated with 95% O$_2$-5% CO2 (pH 7.2–7.4, 290–310 mOsm) at 32°C for 35 min and then allowed to recover at room temperature for an additional 30 min. All recordings were performed at room temperature using an upright microscope (Olympus BX51WI) equipped with differential interference contrast optics and fluorescence imaging (QIACAM fast 1394 with Q-Capture Pro software). Whole-cell patch clamp recordings were obtained from GFP-positive LCIs in the dorsolateral striatum using a MultiClamp 700A Amplifier (Molecular Devices, Sunnyvale, CA) and the pClamp 8.2 software. The patch pipette (3–5 MΩ) contained a cesium-based internal solution (in mM): 125 Cs-methanesulfonate, 4 NaCl, 1 MgCl$_2$, 5 MgATP, 9 EGTA, 8 HEPES, 1 GTP-Tris, 10 phosphocreatine, and 0.1 leupeptin (pH 7.2 with CsOH, 270–280 mOsm) for voltage-clamp recordings or a K-gluconate-based solution containing the following (in mM): 112.5 K-gluconate, 4 NaCl, 17.5 KCl, 0.5 CaCl$_2$, 1 MgCl2, 5 K$_2$ATP, 1 NaGTP, 5 EGTA, 10 HEPES, pH 7.2 (270–280 mOsm/l) for cell attached and current clamp recordings. After breaking through the membrane, cell properties (capacitance, input resistance and time constant) were obtained while holding the membrane potential at −70 mV. Electrode access resistances during all whole cell recordings were maintained at <30 MΩ.

### Spontaneous and evoked postsynaptic currents

Spontaneous PSCs were recorded in gap-free mode, filtered at 1 kHz during acquisition and digitized at 100 μs. sIPSCs were recorded at +20 mV in standard aCSF. Spontaneous sEPSCs were recorded at −70 mV and in the presence of the GABA$_A$ receptor antagonist bicuculline (BIC, 10 μM, Tocris Bioscience, UK). To evoke synaptic currents, a monopolar stimulating electrode (glass-pipette filled with aCSF, impedance ∼1.5 MΩ) was placed in the corpus callosum, 150–200 μm from the recorded cell. QX-314 (4 mM, Tocris Bioscience) was included in the internal pipette solution to block activity-dependent sodium channels, and EPSCs were evoked with cells voltage-clamped at −70 mV in the presence of BIC. Test stimuli (0.5 ms duration) were applied every 20 s at increasing stimulus intensities (0.01–0.1 mA) to assess input–output functions and responses were averaged over three consecutive trials.

### Data analysis

Spontaneous postsynaptic currents were analyzed off-line using the automatic detection protocol within the Mini Analysis Program (Synaptosoft, Decatur, GA) and subsequently checked manually for accuracy. Event analyses were performed blind to genotype. Analyses of individual postsynaptic responses obtained during evoked stimulation and all current clamp measurements were performed using Clampfit 10.2.

## Human postmortem studies

Frozen postmortem putamen samples from three DYT1 patients were provided from University College London. Frozen putamen samples from three control subjects stored at the Michigan Brain Bank were obtained from Dr Roger Albin. Subjects were chosen to control for age, sex, and postmortem interval (see *Table 1*). Small putamen samples were taken with a razor blade, avoiding white matter tracts, and homogenates were prepared as described above for mouse striatum tissue. 10 μg protein lysates were run on 4–20% Biorad gels Mini Protean TGX precast polyacrylamide gels, were transferred to PVDF membranes, and were stained as described above (see *Table 2* for antibody details). Bands were visualized using Supersignal West Pico, Dura, or Femto enhanced chemiluminescence substrates, underwent several exposures to Amersham hyperfilm ECL, and were developed and fixed with an x-ray film developer.

## Statistics

Data are reported as mean ± SEM. Student's t-tests and Chi square tests were performed using Graphpad Prism software (version 6). One-way or two-way ANOVAs were performed using SPSS software (version 22), and post hoc Sidak's or Bonferroni's multiple comparisons tests were performed when significant main effects were observed (p < 0.05). All experiments were repeated at least once before effects were considered significant.

## Acknowledgements

We thank Roger L Albin, Vikram G Shakkottai, and members of the Dauer lab for helpful discussion and critical reviews of the manuscript. We thank Hui Lin Lee, Tom Kennedy, and Livia Rivera for technical support. We thank the laboratory of Dr. Stanley Watson for providing access to their microscope and support for the use of stereoinvestigator and neurolucida software. This research was supported in part by the following grants: RO1NS077730 to WTD (National Institute of Neurological Disorders and Stroke), T32NS007222 to SSP (National Institute of Neurological Disorders and Stroke), UL1TR000433 to OSM (MICHR grant from National Center for Advancing Translational Sciences), R37EB003320 to RTK (National Institute of Biomedical Imaging and Bioengineering).

## Additional information

### Funding

| Funder | Grant reference | Author |
| --- | --- | --- |
| National Institute of Neurological Disorders and Stroke (NINDS) | RO1NS077730 | William T Dauer |
| National Institute of Biomedical Imaging and Bioengineering (NIBIB) | R37EB003320 | Robert T Kennedy |
| National Center for Advancing Translational Sciences (NCATS) | UL1TR000433 | Omar S Mabrouk |
| National Institute of Neurological Disorders and Stroke (NINDS) | T32NS007222 | Samuel S Pappas |

The funders had no role in study design, data collection and interpretation, or the decision to submit the work for publication.

### Author contributions

SSP, Conception and design, Acquisition of data, Analysis and interpretation of data, Drafting or revising the article; KD, SMH, CC, OSM, J-MTW, TMLW, Acquisition of data, Analysis and interpretation of data, Drafting or revising the article; RP, HH, Drafting or revising the article, Contributed unpublished essential data or reagents; RTK, MSL, Analysis and interpretation of data, Drafting or revising the article, Contributed unpublished essential data or reagents; WTD, Conception and design, Analysis and interpretation of data, Drafting or revising the article, Contributed unpublished essential data or reagents

### Ethics

Animal experimentation: All experiments were performed according to the recommendations in the Guide for the Care and Use of Laboratory Animals of the National Institutes of Health. The University of Michigan Committee on the Use and Care of Animals (UCUCA) approved all experiments involving animals (animal use protocol PRO00004330).

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
