## [Decision Letter]

Thank you for submitting your work entitled “Forebrain deletion of the dystonia protein torsinA causes dystonic movements and loss of striatal cholinergic neurons” for peer review at *eLife*. Your submission has been favorably evaluated by a Senior editor and three reviewers, one of whom is a member of our Board of Reviewing Editors.

All individuals responsible for the peer review of your submission have agreed to reveal their identity: Louis Ptáček (Reviewing editor and peer reviewer); Pedro Gonzalez-Alegre and Mark Hallett (peer reviewers).

The reviewers have discussed the reviews with one another, and the Reviewing editor has drafted this decision to help you prepare a revised submission.

No experiments are needed but the authors need to make some revisions to the text to address the points made by the reviewers.

Reviewer #1:

The authors report exciting findings: the association between striatal cholinergic interneurons and dystonia is one that might be predicted based on several previous observations, including (1) cholinergic interneurons are reduced in dystonic hamster model, (2) cholinergic interneurons would be reduced in human benign hereditary chorea (nkx2.1, responsible for migration/development of both cholinergic interneurons and striatal GABAergic interneurons), (3) recent findings that killing cholinergic interneurons reduces threshold for developing tics (Pittenger lab, PNAS, 2015). They focus on the mouse but also add interesting data from human autopsy specimens that support predictions made from the mouse.

The interest in this is of course increased by the fact that prior DYT1 models have had no dystonic phenotype.

One potential criticism is that understanding of mechanism is lacking. The microdialysis is suggestive. The road to basic mechanism is a long and hard one, and this work moves us significantly in that direction. In-depth physiological experiments are necessary but such experiments will be a significant project/manuscript and are beyond the scope of what is reported here.

Reviewer #2:

In this article, Pappas et al. sought to assess the role played by torsinA in forebrain regions as published information supports a role for cholinergic striatal interneurons in the development of dystonia. They describe selective dysfunction and pathology in large cholinergic interneurons (LCI) caused by the loss of the dystonia protein torsinA.

Substantial evidence suggests that DYT1 dystonia is caused by loss of torsinA function. However, the specific cell type (if this is cell-specific) responsible for the appearance of the phenotype remains unknown. Here, the authors developed a new mouse model with conditional deletion of torsinA in embryonic progenitors of cholinergic and GABAergic forebrain neurons. In this model, they encountered dysfunction and loss selectively in LCI following a specific topographic gradient. This pathology was paralleled by the appearance of a specific motor phenotype in mice. Subsequently, they found evidence in human necropsy tissue consistent with this finding (but not conclusive). Overall, the questions asked in this work are significant and relevant to human disease, the article is very well written and the experiments were carried out in a logical order and generated convincing results, with appropriate controls and analysis. The findings represent a significant advance in the understanding of neuronal network in the striatum that control motor activities. This article will be of great interest to the large community of scientists interested on the study of the basal ganglia, and specifically to those focused on dystonia research. In my opinion, this article should be published with minor changes as suggested below.

I believe no additional experimental data is required. The more robust and important contribution of this work is the demonstration of selective dysfunction in LCI upon loss of torsinA within the striatum leading to abnormal motor behavior. The link of this finding to human dystonia is very interesting, but a bit less convincing due to the limitations of the model. Based on that limitation, some of the conclusions might need to be modified for clarity:

The authors claim that, based on their data, selective loss of LCI likely plays a key role in the development of motor dysfunction in DYT1. However, torsinA is a widely expressed protein and they only deleted it here in a subset of neurons. In fact, the same group previously reported (JCI, 2014) the presence of degeneration is other brain areas together with motor dysfunction when conditional deletion was not restricted to the forebrain. That prior data argues that other cell types outside the striatum are equally vulnerable to loss of torsinA function. The way this paper is written, some readers less knowledgeable about the DYT1 field might believe that loss of torsinA leads to selective loss of LCI, and this is not entirely correct as the authors previously showed.

Progressive abnormal clasping behavior has been previously described by Grundman et al. in a rat model of DYT1. The authors mention that this model did not have selective loss of LCI. However, that was not carefully addressed in the DYT1 rats, so that statement is incorrect (we just don't know). It is still possible the rats have a similar LCI phenotype, which would strongly support their findings as expression of the mutated transgene (rather that loss of endogenous torsinA) in the rat model seems to be pan-neuronal.

Abnormal twisting of the trunk during tail hanging is reported in a large proportion of animals, and it is mentioned this “twisting” is specific to their model (though not specifically mentioned, it is implied the way the paper is written that the authors equate this behavior to human dystonia). This is very interesting. This twisting behavior is not included in many rating scales commonly used for clasping and might be simply underreported. This reviewer has observed “twisting” during tail hanging in other models of neurological disease (and in some controls) that exhibit abnormal clasping. I would be cautious about the significance of this “twisting” during tail hanging beyond abnormal motor function linked to LCI pathology.

As the authors properly acknowledge, the significance of the human data is compromised by the scarcity of available tissue and experimental limitations. They obtained 3 DYT1 brains from the UK and three controls from their own brain bank (meaning, different processing, storage, etc of cases and controls). It would have been better to also obtain the controls from the UK to at least diminish this potential confounding factor. There is no available information on the pathology of those brains, and this should be provided if available. Did they also have vascular or Alzheimer pathology, which could affect cholinergic neuron phenotype? (I know of at least one available DYT1 brain with significant AD and vascular pathology). It is implied in the Discussion that the human brains had selective LCI abnormalities. To claim selectivity, the authors would have to look at other brain regions and cell types, as their previous work suggests. I would simply emphasize the limitations of the human data a bit more, including toning down the sentence about LCI selectivity in human tissue in the Abstract.

Reviewer #3:

Using a model of the Dlx-CKO mouse, the authors have analyzed cholinergic function and its relationship to behavioral abnormalities with resemblance to human dystonia. In a detailed series of experiments, they show that there is a developmental abnormality of LCI that correlates directly with the abnormal behavior. They show that the abnormality is specific for the cholinergic system, that some cells are lost and surviving cells are malfunctioning. They also show that anticholinergic medication improves the behavior, similar to human patients, and that human putamina also have a cholinergic abnormality.

The work is remarkable for its thoroughness and clarity of exposition. The final story relates the animal model and human disease on many levels, and is a major advance in understanding dystonia.

The authors should probably quote the McNaught paper of DYT1 pathology that found inclusions in the cholinergic neurons (even though they did not find any pathology in the striatum nor any cell loss).

I see no need for revision, but do have a few questions that they might want to try to answer or address.

1) The genetic abnormality is in GABA neurons as well as cholinergic neurons. Why aren't they involved?

2) The histological analysis shows the cholinergic abnormality mostly in the motor part of the striatum. Are there any observations about a distribution relating to the patch-matrix histology of the striatum? This histology was first defined on cholinergic markers.

3) Based on the cell biology of torsinA, how does the abnormality lead to the damage of the cholinergic neuron? The cell biology must be cell specific in some way.

4) Is there any explanation for the timing of the effect? Why is there the delay before the pathological process begins? And, why does it stop and plateau?

---

## [Author Response]

Reviewer #1:

*The authors report exciting findings: the association between striatal cholinergic interneurons and dystonia is one that might be predicted based on several previous observations, including (1) cholinergic interneurons are reduced in dystonic hamster model, (2) cholinergic interneurons would be reduced in human benign hereditary chorea (nkx2.1, responsible for migration/development of both cholinergic interneurons and striatal GABAergic interneurons), (3) recent findings that killing cholinergic interneurons reduces threshold for developing tics (Pittenger lab, PNAS 2015). They focus on the mouse but also add interesting data from human autopsy specimens that support predictions made from the mouse*.

*The interest in this is of course increased by the fact that prior DYT1 models have had no dystonic phenotype*.

One potential criticism is that understanding of mechanism is lacking. The microdialysis is suggestive. The road to basic mechanism is a long and hard one, and this work moves us significantly in that direction. In-depth physiological experiments are necessary but such experiments will be a significant project/manuscript and are beyond the scope of what is reported here.

We are grateful to Reviewer 1 for recognizing the strength of our findings. We agree that identifying the mechanisms linking striatal dysfunction to abnormal movement are an exciting “next step” of this work, and are we are currently pursuing electrophysiological studies to explore this issue.

Reviewer #2:

*The authors claim that, based on their data, selective loss of LCI likely plays a key role in the development of motor dysfunction in DYT1. However, torsinA is a widely expressed protein and they only deleted it here in a subset of neurons. In fact, the same group previously reported (JCI, 2014) the presence of degeneration is other brain areas together with motor dysfunction when conditional deletion was not restricted to the forebrain. That prior data argues that other cell types outside the striatum are equally vulnerable to loss of torsinA function. The way this paper is written, some readers less knowledgeable about the DYT1 field might believe that loss of torsinA leads to selective loss of LCI, and this is not entirely correct as the authors previously showed*.

We agree that our initial manuscript may have given the impression that LCI vulnerability is unique throughout the brain, rather than within the striatum. In our revised manuscript we clarify this point. In the Discussion, we now state that “our studies in postmortem striatum from DYT1 subjects, the first to examine cell type-specific markers in this context, are consistent with our experimental findings pointing to selective cholinergic defect in this structure. Previously work in overtly symptomatic DYT1 models demonstrates loss of discrete cell types in other motor-related structures (e.g., loss of neurons of the deep cerebellar nuclei, with preservation of all other cerebellar cell types; [28]).” We have made several other changes to highlight that the selectivity observed here is within the striatum (all marked with “track changes” in the revised manuscript).

*Progressive abnormal clasping behavior has been previously described by Grundman et al. in a rat model of DYT1. The authors mention that this model did not have selective loss of LCI. However, that was not carefully addressed in the DYT1 rats, so that statement is incorrect (we just don't know). It is still possible the rats have a similar LCI phenotype, which would strongly support their findings as expression of the mutated transgene (rather that loss of endogenous torsinA) in the rat model seems to be pan-neuronal*.

We appreciate this correction. We have corrected and clarified this point in the Discussion point by stating that “abnormal clasping is present in a transgenic rat model of dystonia, but it is not known if striatal LCI number is reduced in these animals (17).”

*Abnormal twisting of the trunk during tail hanging is reported in a large proportion of animals, and it is mentioned this* “*twisting*” *is specific to their model (though not specifically mentioned, it is implied the way the paper is written that the authors equate this behavior to human dystonia). This is very interesting. This twisting behavior is not included in many rating scales commonly used for clasping and might be simply underreported. This reviewer has observed* “*twisting*” *during tail hanging in other models of neurological disease (and in some controls) that exhibit abnormal clasping. I would be cautious about the significance of this* “*twisting*” *during tail hanging beyond abnormal motor function linked to LCI pathology*.

We appreciate Reviewer 2 sharing these observations. As noted in the Methods section, our behavioral assessments were performed by several independent observers, all of whom were blinded to genotype. This analysis showed brief twisting episodes in only a very small number of control animals. In our pharmacological studies, only severe twisting episodes were considered abnormal and were scored accordingly. We agree that the relationship between twisting during tail suspension and human dystonia is unclear, and we have clarified this point in several instances in our revised manuscript, including by modifying the Title.

*As the authors properly acknowledge, the significance of the human data is compromised by the scarcity of available tissue and experimental limitations. They obtained 3 DYT1 brains from the UK and three controls from their own brain bank (meaning, different processing, storage, etc of cases and controls). It would have been better to also obtain the controls from the UK to at least diminish this potential confounding factor. There is no available information on the pathology of those brains, and this should be provided if available. Did they also have vascular or Alzheimer pathology, which could affect cholinergic neuron phenotype? (I know of at least one available DYT1 brain with significant AD and vascular pathology). It is implied in the Discussion that the human brains had selective LCI abnormalities. To claim selectivity, the authors would have to look at other brain regions and cell types, as their previous work suggests. I would simply emphasize the limitations of the human data a bit more, including toning down the sentence about LCI selectivity in human tissue in the Abstract*.

We agree entirely with these sentiments. We have added additional information available to us regarding the postmortem tissue in the “Human Subject Data” table (Figure 8–figure supplement 1). This information shows that the subjects did not exhibit known vascular or AD pathology.

To more accurately report the results of our human postmortem studies, we removed the term ‘selective’ from the Abstract and several portions of the Discussion, and have clarified other statements regarding cholinergic selectivity. We include the following statement in our revised manuscript: “Our analysis of postmortem tissue was restricted to striatum; future work is required to determine whether there is cell loss in extra-striatal areas corresponding to those identified in murine models.”

Reviewer #3:

The authors should probably quote the McNaught paper of DYT1 pathology that found inclusions in the cholinergic neurons (even though they did not find any pathology in the striatum nor any cell loss).

In our revised manuscript, we have inserted this citation into the Discussion regarding the human DYT1 tissue analysis.

*I see no need for revision, but do have a few questions that they might want to try to answer or address*.

1) The genetic abnormality is in GABA neurons as well as cholinergic neurons. Why aren't they involved?

Considering how early torsinA is deleted, it is remarkable that forebrain GABA neuron number morphological appearance are normal. As we point out in the revised manuscript, they may be functionally abnormal, yet conditional deletion of torsinA from these cells does not cause overt abnormalities (58). We are pursuing electrophysiological studies to address this question. We agree with the reviewer that the selectivity of cell loss is quite remarkable. While we do not yet have an answer to this question, the mechanisms of selective vulnerability is one we are actively exploring in a range of molecular and related studies of vulnerable and invulnerable neural cell types.

2) The histological analysis shows the cholinergic abnormality mostly in the motor part of the striatum. Are there any observations about a distribution relating to the patch-matrix histology of the striatum? This histology was first defined on cholinergic markers.

We have explored this question but do not observe any topographic loss according to patch/matrix organization. We note this point in our revised manuscript.

*3) Based on the cell biology of torsinA, how does the abnormality lead to the damage of the cholinergic neuron? The cell biology must be cell specific in some way*.

We agree completely that the selectivity of LCI loss must reflect a key aspect of the cellular function of torsinA to which these cells are uniquely sensitive (within striatum). Indeed, the identification of torsinA-sensitive neural classes throughout the CNS is motivated by the expectation that understanding the mechanistic basis of this vulnerability will reveal key aspects of torsinA biology critical to disease pathogenesis. As noted above, we are currently pursuing studies in defined vulnerable and invulnerable cell types to address this question. Our understanding of the cellular role of torsinA is still quite limited, but abnormal protein quality control of a molecule particularly important to LCI is one possibility. TorsinA appears to play a role in protein quality control in the ER. To hint at this possibility, our revised manuscript now states, in the Discussion, that “postmortem analyses of DYT1 subjects suggests the presence of inclusion bodies in pedunculopontine cholinergic neurons, but did not report cell loss ([31]; #3647). These observations, and additional work ([28] and reviewed by ; [10]), suggest that abnormal protein quality control may represent a pathogenic event in DYT1 dystonia.”

4) Is there any explanation for the timing of the effect? Why is there the delay before the pathological process begins? And, why does it stop and plateau?

We agree that the mechanism underlying the neurodevelopmental timing is LCI loss is another intriguing question. This timing coincides with the period of striatal circuit maturation when afferent connections are established and projection neurons and interneurons become physiologically active and mature (18; 53; 27). Our working hypothesis is that this period of circuit maturation represents an important critical period for torsinA function. This is also suggested by previous models. We inserted the following statement into the Discussion to address this important point: “Previous torsinA loss-of-function models also demonstrate cell loss restricted to a neurodevelopmental window (28). […] cell loss and behavior onset during a neurodevelopmental window periods is consistent with a critical period for torsinA function in CNS development.”